# CoLA: Co-Calibrated Logit Adjustment for Long-Tailed Semi-Supervised Learning

**Qian Shao[1] & Qiyuan Chen[1] & Jiahe Chen[1] & Zepeng Li[3] & Qianqian Tang[4] & Hongxia Xu[2]\* & Jian Wu[5]\***

[1]Transvascular Implantation Devices Research Institute and College of Computer Science & Technology, Zhejiang University, Hangzhou, China
[2]Liangzhu Laboratory and WeDoctor Cloud, Hangzhou, China
[3]The State Key Laboratory of Blockchain and Data Security, Zhejiang University
[4]Shaoxing Tangtang Technology Co., Ltd.
[5]Zhejiang Key Laboratory of Medical Imaging Artificial Intelligence, Hangzhou, China
{qianshao, einstein, wujian2000}@zju.edu.cn

## ABSTRACT

Long-tailed semi-supervised learning is hampered by a vicious cycle of confirmation bias, where skewed pseudo-labeling progressively marginalizes tail classes. This challenge is compounded in real-world scenarios by a class distribution mismatch between labeled and unlabeled data, rendering the bias unpredictable and difficult to mitigate. While existing methods adapt Logit Adjustment (LA) using dynamic estimates of the unlabeled distribution, we argue their effectiveness is undermined by two critical limitations stemming from LA's core design, *i.e.*, its class-wise and overall adjustment mechanisms. First, their reliance on simple frequency counting overestimates the prevalence of head classes due to sample redundancy, leading to harmful over-suppression. Second, and more critically, they overlook the interplay between the above two types of adjustment, treating the overall adjustment strength as a fixed hyperparameter. This is a significant oversight, as we empirically find that the optimal strength is highly sensitive to the estimated distribution. To address these limitations, we propose Co-Calibrated Logit Adjustment (CoLA), a framework that co-designs the class-wise and overall LA components. Specifically, CoLA refines the class-wise adjustment by estimating each class's effective sample size via the effective rank of its representations. Subsequently, it formulates the overall adjustment strength as a learnable parameter, which is optimized through a meta-learning procedure on a proxy validation set constructed to mirror the refined distribution. Supported by a theoretical generalization bound, our extensive experiments show that CoLA outperforms existing baselines on $4$ public benchmarks across standard long-tail setups.

## 1 INTRODUCTION

Semi-Supervised Learning (SSL) (Laine & Aila, 2017; Sohn et al., 2020; Nguyen, 2024) often struggles in real-world applications due to the inherent long-tailed nature of data, a critical problem known as Long-Tailed Semi-Supervised Learning (LTSSL) (Kim et al., 2020; Lee et al., 2021; Wei et al., 2021). In the LTSSL setting, a model initially trained on skewed labeled data develops a strong bias towards the majority classes. This initial bias is then amplified when the model generates a large volume of equally biased pseudo-labels from unlabeled data. This process creates a vicious cycle of confirmation bias, causing the model to become increasingly overconfident in majority classes while progressively neglecting the tail classes (Duan et al., 2022; Fan et al., 2022; Guo & Li, 2022). Consequently, the ability to produce high-quality, unbiased pseudo-labels has become the paramount challenge in LTSSL (Wei & Gan, 2023; Ma et al., 2024; Xing et al., 2025).

Recent advancements in LTSSL predominantly rely on Logit Adjustment (LA) strategy (Menon et al., 2021) or its variants to enhance the quality of pseudo-labels (Wei & Gan, 2023; Ma et al.,

---

\*Corresponding author.

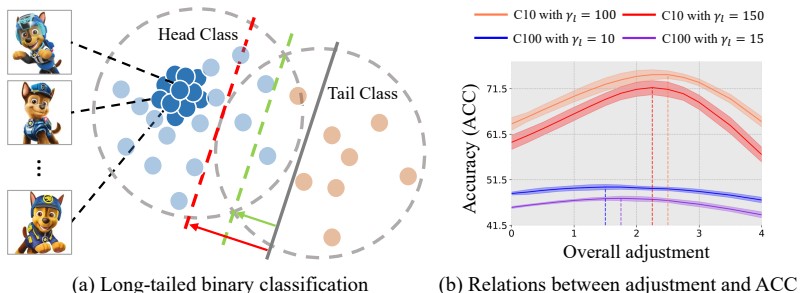

(a) Long-tailed binary classification     (b) Relations between adjustment and ACC

Figure 1: **(a)** An unadjusted classifier is biased towards head classes (grey solid line), which naively uses class frequency for class-wise adjustment, leading to over-suppression (red dashed line). This is because frequency counting ignores sample redundancy (dark blue spheres), inflating the estimated importance of head classes. De-duplication could mitigate this issue (green dashed line). **(b)** The optimal overall adjustment strength (four dashed lines) is highly sensitive to the distribution and the number of classes. Counter-intuitively, it does not always correlate positively with the imbalance ratio $\gamma_l$. For instance, on CIFAR-10-LT (C10), the optimal strength for $\gamma_l = 100$ is greater than for $\gamma_l = 150$, highlighting the difficulty of relying on a pre-defined strength in complex scenarios.

2024; Hou & Jia, 2025). At its core, LA comprises two key components: a class-wise adjustment, which controls the relative adjustment strength of logits for different classes according to the prior probability (*i.e.*, suppressing the head class and encouraging the tail class), and an overall adjustment, which uniformly controls the overall scaling magnitude. However, since the true distribution of unlabeled data is unknown, achieving an accurate adjustment is challenging. Consequently, many existing methods (Ma et al., 2024; Hou & Jia, 2025) resort to using a pre-defined anchor distribution as a proxy for the true class priors. This static assumption proves brittle in more realistic and complex scenarios (Du et al., 2024), where the actual class distribution inevitably deviates from the pre-defined anchor, rendering these methods ineffective.

More sophisticated works (Wei & Gan, 2023; Li et al., 2024) have pursued dynamically estimating the class distribution of unlabeled data. Despite their conceptual advantages, these approaches encounter a critical two-fold dilemma. First, they typically estimate class distributions via a naive frequency-counting of high-confidence predictions. This process often overlooks sample redundancy, an issue particularly pronounced in head classes that are saturated with visually similar instances (Cui et al., 2019; Zhang et al., 2024). Thus, the estimated prevalence of these classes becomes inflated, leading to an over-suppression problem that ultimately degrades model performance (Figure 1a). Furthermore, these approaches often ignore the interplay between the two adjustments, treating the overall adjustment strength as a fixed hyperparameter. In contrast, we contend that an accurate class-wise adjustment is a prerequisite for determining the optimal overall adjustment. Our empirical analysis reveals that the optimal overall adjustment is highly sensitive to the estimated class distribution and the number of classes (Figure 1b). This exposes the second facet of the dilemma: their reliance on a pre-defined overall strength, rendering them incapable of adapting to varying dataset characteristics and ultimately degrading the quality of pseudo-labels. Moreover, most LA-based works lack theoretical grounding to guide the selection of its optimal value.

To address these issues, we propose Co-Calibrated Logit Adjustment (CoLA), which co-designs the class-wise and overall adjustment of LA for any given class distribution. For accurate class-wise adjustment, we introduce De-Duplicated Distribution Estimation (DDDE) to tackle the over-suppression problem. Departing from naive frequency counting, DDDE quantifies sample redundancy by calculating the effective rank (Roy & Vetterli, 2007) of each class's representations, which is inspired by the concept of effective sample size (Cui et al., 2019). This process yields a more robust and accurate estimate of class prevalence. Based on the estimated distribution, we design Logit Meta-Calibration (LMC), which first constructs a proxy validation set by resampling labeled data to match the estimated distribution, and then meta-learns the optimal strength on this proxy set.

Theoretically, we establish a generalization bound for our meta-learning procedure and provide a convexity analysis of the optimization objective. Empirically, extensive experiments demonstrate that CoLA achieves new state-of-the-art (SOTA) performance on standard long-tailed benchmarks. Furthermore, comprehensive ablation and visualization experiments validate the individual contributions of the key components. The main contributions of this article are as follows:

- We reveal the critical interplay within the LA mechanism, and propose CoLA that adaptively co-designs the class-wise and overall adjustment of LA for LTSSL;

- We introduce DDDE, which mitigates the over-suppression problem of head classes by leveraging a redundancy-aware estimate of the unlabeled class distribution;

- We design LMC, a meta-learning strategy that learns the optimal overall adjustment strength on a proxy set, which mirrors the characteristics of the unlabeled set;

- We validate our proposed CoLA rigorously through theoretical analysis and extensive experiments on 4 benchmarks across 6 different distributions.

## 2 RELATED WORK

LTSSL has emerged to address the critical challenge of performance degradation in conventional SSL methods (Lee et al., 2013; Sohn et al., 2020) when confronted with class-imbalanced data. LTSSL approaches often involve LA (Li et al., 2024; Xing et al., 2025), loss re-weighting (Lai et al., 2022; Wang et al., 2022a), re-sampling (Lee et al., 2021; Wei et al., 2021), data augmentation via mixing (Fan et al., 2022; Zheng et al., 2024), distribution alignment (Kim et al., 2020; Lazarow et al., 2023; Sanchez Aimar et al., 2024), and others (Kim et al., 2025; Oh et al., 2022).

Due to its efficacy and ease of implementation, LA (Menon et al., 2021) is widely used in recent research, including post-hoc correction and/or loss function modification. For instance, CDMAD (Lee & Kim, 2024) employs the post-hoc LA approach, which derives the adjustment strength from the classifier's biased degree by calculating logits on solid color images. LCGC (Xing et al., 2025) further provides a theoretical foundation for this methodology, enhancing its robustness. In a different vein, TCBC (Li et al., 2024) corrects the model bias and refines the pseudo-labels by modifying the LA-based loss function. CPE (Ma et al., 2024) designs three distinct classifiers, each tailored to a pre-defined anchor distribution through a specialized LA-based loss function. Building on this, Meta-Expert (Hou & Jia, 2025) introduces a dynamic gating mechanism to select the most suitable classifier for each unlabeled instance, aiming for more accurate pseudo-label generation. In addition, ACR (Wei & Gan, 2023) employs a dual-branch structure that uses these two LA approaches respectively, where the standard branch calculates the distance to three pre-defined anchor distributions to determine the strength of post-hoc LA for accurate pseudo labels, while the balanced branch utilizes an LA-based loss function to generate class-balanced pseudo-labels.

Although the above methods have demonstrated considerable success, most of them lie in a shared reliance on a small, discrete set of pre-defined anchor distributions, which fail to handle class distributions that fall outside these anchors. Consequently, their applicability is limited in more complex and realistic scenarios where distribution shifts can be arbitrary and unpredictable (Du et al., 2024). Related work on other types of LTSSL algorithms can be found in Appendix B.

## 3 PRELIMINARY

Let $\mathcal{X}$ be the input space and $\mathcal{Y} = \{1, 2, ..., K\}$ be the label space, where $K$ is the number of classes. We consider two underlying joint distributions over $\mathcal{X} \times \mathcal{Y}$: a labeled data distribution $P_{\mathbf{X}_l, Y_l}(\boldsymbol{x}, y)$, and an unlabeled data distribution $P_{\mathbf{X}_u, Y_u}(\boldsymbol{x}, y)$, abbreviated as $P_l$ and $P_u$, respectively. The training data consists of two distinct sets: (1) a labeled set $\mathcal{D}_l = \{(\boldsymbol{x}_i^l, y_i)\}_{i=1}^N$ drawn $i.i.d.$ from $P_l$; (2) an unlabeled set $\mathcal{D}_u = \{\boldsymbol{x}_j^u\}_{j=1}^M$, where $N$ and $M$ are the numbers of samples in the labeled and unlabeled sets, respectively. For the purpose of analysis, we posit that each sample $\boldsymbol{x}_j^u$ is associated with a true but unobserved label $y_j^u$, such that the complete pair $(\boldsymbol{x}_j^u, y_j^u)$ is drawn $i.i.d.$ from $P_u$. Crucially, only $\{\boldsymbol{x}_j^u\}_{j=1}^M$ are accessible during the training phase.

For each class $k \in \mathcal{Y}$, let $N_k$ be the number of samples in $\mathcal{D}_l$. We order the classes such that $N_1 \geq N_2 \geq \cdots \geq N_K$. The imbalance ratio of the labeled set is defined as $\gamma_l = \frac{N_1}{N_K}$, and it is assumed that that $\gamma_l \gg 1$. Similarly, let $M_k$ represent the true (unobserved) number of samples for class $k$ within the unlabeled dataset $\mathcal{D}_u$. We denote the unlabeled class indices with the largest and smallest sample numbers, and the imbalance ratio as $k_{\max}$, $k_{\min}$, and $\gamma_u = M_{k_{\max}}/M_{k_{\min}}$, respectively. These per-class counts $\{M_k\}_{k=1}^K$ and the ratio $\gamma_u$ are unknown during training.

Following the previous works (Du et al., 2024), we consider six distributions of unlabeled data, which cover various real-world situations: (1) consistent long-tailed distribution, with $(\gamma_u, k_{\max}, k_{\min}) = (\gamma_l, 1, K)$; (2) uniform distribution, with $\gamma_u = 1$; (3) reversed distribution, with $(\gamma_u, k_{\max}, k_{\min}) = (\gamma_l, K, 1)$; (4) middle distribution, with $(\gamma_u, k_{\max}, k_{\min}) = (\gamma_l, \lceil \frac{K}{2} \rceil, K^{(K+1) \bmod 2})$; (5) head-tail distribution, with $(\gamma_u, k_{\max}, k_{\min}) = (\gamma_l, K^{(K+1) \bmod 2}, \lceil \frac{K}{2} \rceil)$; (6) unknown distribution, which means that the dataset itself contains some unlabeled data, such as STL-10 (Coates et al., 2011).

The goal of LTSSL is to learn a deep model on the train set $\mathcal{D}_{\text{train}} = \mathcal{D}_l \cup \mathcal{D}_u$ that achieves good performances on a class-balanced test set $\mathcal{D}_{\text{test}}$. Let $z(\boldsymbol{x})$ denote the backbone of the deep model, transforming the sample $\boldsymbol{x} \in \mathcal{D}_{\text{train}}$ into the output logits. Following previous works (Wang et al., 2022a; Wei & Gan, 2023), we adopt FixMatch (Sohn et al., 2020) as a backbone SSL algorithm. For a labeled sample, FixMatch computes the cross-entropy (CE) loss $\mathcal{L}_{\text{CE}}$ between the true label and the model's prediction. For an unlabeled sample, FixMatch first performs strong (Cubuk et al., 2020; DeVries, 2017) and weak data augmentation on it separately, denoted by $\mathcal{A}(\boldsymbol{x}^u)$ and $\alpha(\boldsymbol{x}^u)$. Then, FixMatch predicts the class probabilities for them as $\sigma(z(\mathcal{A}(\boldsymbol{x}^u)))$ and $\sigma(z(\alpha(\boldsymbol{x}^u)))$, respectively, where $\sigma(\cdot)$ denotes the softmax function. The total loss of the backbone $\mathcal{L}$ is defined by

$$\mathcal{L} = \frac{1}{N} \sum_{i=1}^{N} \mathcal{L}_{\text{CE}} \left( y_i, \sigma(z(\alpha(\boldsymbol{x}_i^l))) \right) + \frac{1}{M} \sum_{j=1}^{M} \mathbb{I}(\|\sigma(z(\alpha(\boldsymbol{x}_j^u)))\|_\infty > \rho) \cdot \mathcal{L}_{\text{CE}} \left( \hat{y}_j, \sigma(z(\mathcal{A}(\boldsymbol{x}_j^u))) \right),$$

where $\mathbb{I}(\cdot)$ is an indicator function, with threshold $\rho$, and $\hat{y}_j = \arg\max_{y \in \mathcal{Y}} z(\alpha(\boldsymbol{x}_j^u))_y$ denotes the pseudo label. However, training with the objective $\mathcal{L}$ on a long-tailed set causes the model to develop a significant bias towards majority classes, which directly compromises the quality of pseudo-labels and exacerbates the confirmation bias cycle. To counteract this effect, the post-hoc LA strategy (Menon et al., 2021) is widely adopted to generate accurate pseudo labels via re-calibrating the predictions on unlabeled data:

$$\hat{y}_j = \arg\max_{y \in \mathcal{Y}} \left( z(\alpha(\boldsymbol{x}_j^u))_y - \tau \cdot \log \hat{P}_{Y_u}(y) \right), \tag{1}$$

where $\hat{P}_{Y_u}(y)$ are estimates of the unlabeled class distribution $P_{Y_u}(y)$ which controls the strength of class-wise adjustment, and $\tau > 0$ is a hyperparameter that controls the strength of overall adjustment. This process effectively penalizes the logits of classes presumed to be frequent under $\hat{P}_{Y_u}(y)$, thereby increasing the likelihood of assigning pseudo-labels to minority classes.

## 4 METHODOLOGY

In this section, we first introduce DDDE method for estimating the class distribution of unlabeled data in Section 4.1, which is specifically designed to handle sample redundancy. With this accurate prior, we then propose LMC in Section 4.2, a data-driven strategy that meta-learns the optimal overall adjustment strength on a distribution-matched proxy set. Finally, Section 4.3 elaborates on the integration of these components into an end-to-end training pipeline.

### 4.1 DE-DUPLICATED DISTRIBUTION ESTIMATION

To solve the over-suppression problem caused by the overestimation of the prevalence of head classes, we draw inspiration from the concept of Effective Number (EN) of samples (Cui et al., 2019), and propose the DDDE method to better estimate class frequencies. Specifically, we quantify the EN of samples using the effective rank (erank) (Roy & Vetterli, 2007).

For each class $y \in \mathcal{Y}$, we first gather the representations $\{\boldsymbol{z}_j^y\}_{j=1}^{m_y} \in \mathbb{R}^d$ of samples $\boldsymbol{x}_j^u$ whose pseudo-label is $y$ and confidence score $\|\sigma(z(\alpha(\boldsymbol{x}_j^u)))\|_\infty$ is above the threshold $\rho$, then forming a feature matrix $\boldsymbol{Z}_y = (\boldsymbol{z}_1^y, \boldsymbol{z}_2^y, ..., \boldsymbol{z}_{m_y}^y) \in \mathbb{R}^{d \times m_y}$, where $m_y$ and $d$ are the number and the dimension of representations, respectively. We assume that $\boldsymbol{Z}_y$ is full-rank, since it is exceedingly rare for any two representation vectors to be perfectly linearly dependent, and then we compute its singular values, $s_1, s_2, ..., s_{m_y}$. These values represent the energy of the data along its principal components. To formally apply Shannon entropy, we interpret the normalized singular value spectrum within a

probabilistic framework. Specifically, let $\mathcal{K}$ be a discrete random variable representing the index of a principal component within the feature space. We define the probability distribution of $\mathcal{K}$ as $p(i) = \mathbb{P}(\mathcal{K} = i) = s_i / \sum_{j=1}^{m_y} s_j$. Physically, $p(i)$ represents the probability that a unit of feature variance is aligned with the $i$-th principal component. The erank is then defined as the exponential of the Shannon entropy of this distribution:

$$\text{erank}(\boldsymbol{Z}_y) = \exp\left(-\sum_{i=1}^{m_y} p(i) \log p(i)\right).$$

The erank effectively measures the uncertainty of the energy distribution across principal directions, serving as a robust proxy for the EN of samples. Finally, the de-duplicated class distribution of the unlabeled data, $\hat{P}_{Y_u}(y)$, is obtained by normalizing the effective counts across all classes:

$$\hat{P}_{Y_u}(y) = \frac{\text{erank}(\boldsymbol{Z}_y)}{\sum_{k \in \mathcal{Y}} \text{erank}(\boldsymbol{Z}_k)}.$$

## 4.2 Logit Meta-Calibration

As shown in (1), the post-hoc LA technique introduces a hyperparameter $\tau$ that modulates the overall adjustment strength. While empirical results demonstrate its effectiveness, the optimal choice of $\tau$ is data-dependent, and no theoretical guidance is provided for its selection. To address this limitation, we propose a principled, data-driven approach LMC to determine $\tau$ by making it a learnable parameter, specifically tailoring it for pseudo-label generation in the context of LTSSL. First, we construct a proxy set $\mathcal{D}_v$ by resampling from the labeled set $\mathcal{D}_l$ to match the estimated unlabeled distribution $\hat{P}_{Y_u}(y)$. Specifically, for each labeled sample $(\boldsymbol{x}_i^l, y_i) \in \mathcal{D}_l$, we assign a selection probability:

$$\mathbb{P}((\boldsymbol{x}_i^l, y_i) \text{ is selected}) = \frac{\hat{P}_{Y_u}(y_i)}{N_{y_i}} \Big/ \max_{y \in \mathcal{Y}}\left(\frac{\hat{P}_{Y_u}(y)}{N_y}\right),$$

where the denominator $\max_{y \in \mathcal{Y}}(\hat{P}_{Y_u}(y)/N_y)$ serves as a normalization to ensure all probabilities are at most 1, thus implementing a form of rejection sampling to prevent oversampling. After performing a Bernoulli trial for each sample with its corresponding probability, we obtain $\mathcal{D}_v = \{(\boldsymbol{x}_i^v, y_i^v)\}_{i=1}^V$, and optimize $\tau$ by minimizing the CE loss on $\mathcal{D}_v$. The optimal $\tau^*$ is found by:

$$\tau^* = \arg\min_\tau \frac{1}{V} \sum_{i=1}^V \mathcal{L}_{\text{CE}}\Big(y_i^v, \sigma(z(\alpha(\boldsymbol{x}_i^v)) - \tau \cdot \boldsymbol{p})\Big),$$

where $\boldsymbol{p} = (\hat{P}_{Y_u}(1), \hat{P}_{Y_u}(2), ..., \hat{P}_{Y_u}(K))$ is the vector of our estimated de-duplicated class frequencies. Notably, we use LA term $-\tau \cdot \boldsymbol{p}$, which is motivated by theoretical insights from (Mor & Carmon, 2025) and deviates from the original post-hoc LA, which uses the logarithm of the class frequencies. This linear term, as opposed to the logarithmic one, avoids potential numerical instability and overly aggressive penalization for classes with very small estimated probabilities, leading to a more stable optimization process. Finally, $\tau^*$ is used to calibrate the logits for generating pseudo-labels on the unlabeled data, thereby producing higher-quality supervisory signals.

## 4.3 End-to-End Training

Following the settings of previous researches (Wei & Gan, 2023; Lee et al., 2021; Park et al., 2024), we adopt a dual-branch architecture comprising a balanced branch and a standard branch. The balanced branch, on which we apply DDDE, is designed to produce class-balanced predictions. Concurrently, the standard branch leverages LMC to generate high-quality pseudo-labels. The overall adjustment parameter $\tau$ is managed in a two-stage process. During an initial warm-up phase, $\tau$ is configured according to ACR (Wei & Gan, 2023). Once the model achieves a reliable estimate of the class distribution, LMC learns the optimal $\tau$ for the subsequent training process. Additional implementation details and time complexity analysis are provided in Appendix G.2 and H, respectively.

## 5 GENERALIZATION ANALYSIS

We provide a theoretical analysis of the generalization performance of a classifier parameterized by $\tau$. Our goal is to bound the expected risk on the target distribution $P_u$ for a classifier $h_\tau$ whose controlling parameter $\tau$ is optimized on the proxy dataset $\mathcal{D}_v$. Tighter bounds imply higher quality pseudo-labels. Let $\mathcal{H}_\tau$ be a hypothesis space indexed by $\tau$, where each $h_\tau : \mathcal{X} \to \mathcal{Y}$ is a classifier. Let $\ell(h_\tau(\boldsymbol{x}), y) : \mathcal{Y} \times \mathcal{Y} \to \mathbb{R}^+$ be a loss function. Then we make four assumptions as follows.

**Assumption 1** (Lipschitz and Bounded Loss). *The loss function $\ell(h_\tau(\boldsymbol{x}), y)$ is L-Lipschitz with respect to $h_\tau(\boldsymbol{x})$ (the classifier's output) for all $y \in \mathcal{Y}$ and is upper-bounded by $U$.*

Assumption 1 is a standard assumption satisfied by many common loss functions like CE loss on a compact domain.

**Assumption 2.** *Each sample $(\boldsymbol{x}_i^v, y_i^v)$ in $\mathcal{D}_v$ is drawn i.i.d. from an underlying joint distribution $P_{\boldsymbol{X}_v, Y_v}(\boldsymbol{x}, y)$, abbreviated as $P_v$, and its marginal label distribution satisfies $P_{Y_v}(y) = \hat{P}_{Y_u}(y)$.*

While $\mathcal{D}_v$ is practically sampled from the finite set $\mathcal{D}_l$, for the purpose of theoretical analysis, we model it as being drawn i.i.d. from an underlying distribution $P_v$, which is a standard approach in learning theory. The latter part of Assumption 2 holds by construction within our framework. Specifically, the resampling procedure for constructing $\mathcal{D}_v$ ensures that the expected marginal label distribution of the resulting dataset is equal to $\hat{P}_{Y_u}(y)$.

**Assumption 3.** *The labeled and unlabeled data share the same class-conditional distribution.*

Assumption 3 is a central and common assumption in SSL, i.e., $P_{\boldsymbol{X}_u|Y_u}(\boldsymbol{x}|y) = P_{\boldsymbol{X}_l|Y_l}(\boldsymbol{x}|y)$. Since the validation set $\mathcal{D}_v$ is constructed via rejection sampling from the labeled set $\mathcal{D}_l$, they inherently share the same class-conditional distribution, implying $P_{\boldsymbol{X}_v|Y_v}(\boldsymbol{x}|y) = P_{\boldsymbol{X}_l|Y_l}(\boldsymbol{x}|y)$. Following the literature on covariate and distribution shift (Shimodaira, 2000), we introduce an importance weight function $w(\boldsymbol{x}, y) = \frac{P_{\boldsymbol{X}_u, Y_u}(\boldsymbol{x}, y)}{P_{\boldsymbol{X}_v, Y_v}(\boldsymbol{x}, y)}$. Under Assumption 3, the weight simplifies to the function $w(\boldsymbol{x}, y) = \frac{P_{\boldsymbol{X}_u|Y_u}(\boldsymbol{x}|y) \cdot P_{Y_u}(y)}{P_{\boldsymbol{X}_v|Y_v}(\boldsymbol{x}|y) \cdot P_{Y_v}(y)} = \frac{P_{\boldsymbol{X}_l|Y_l}(\boldsymbol{x}|y) \cdot P_{Y_u}(y)}{P_{\boldsymbol{X}_v|Y_v}(\boldsymbol{x}|y) \cdot P_{Y_v}(y)} = \frac{P_{Y_u}(y)}{P_{Y_v}(y)}$, denoted as $w(y)$.

**Assumption 4.** *The importance weight $w(y)$ is upper-bounded by a constant $B \geq 1$.*

Assumption 4 holds provided that for all classes $y \in \mathcal{Y}$, our estimated class frequency $\hat{P}_{Y_u}(y)$ is not a severe underestimate of its ground-truth value $P_{Y_u}(y)$. This assumption highlights the importance of our proposed DDDE, which requires that our method does not produce substantial underestimates of the true class proportions for the bound to hold.

The expected risk of $h_\tau$ on the target distribution $P_u$ is $R_{P_u}(h_\tau) = \mathbb{E}_{(\boldsymbol{X}_u, Y_u) \sim P_u}[\ell(h_\tau(\boldsymbol{X}_u), Y_u)]$. The empirical risk on the proxy dataset $\mathcal{D}_v$ is $\hat{R}_{\mathcal{D}_v}(h_\tau) = \frac{1}{V} \sum_{i=1}^{V} \ell(h_\tau(\boldsymbol{x}_i^v), y_i^v)$. The weighted empirical risk is defined as $\hat{R}_{\mathcal{D}_v, w}(h_\tau) = \frac{1}{V} \sum_{i=1}^{V} w(y_i^v) \ell(h_\tau(\boldsymbol{x}_i^v), y_i^v)$. We can now state our generalization bound for $h_\tau$ on $P_u$.

**Proposition 1.** *Under Assumptions 1, 2, 3, and 4, for any $\delta \in (0, 1)$, with probability at least $1 - \delta$ over the draw of $\mathcal{D}_v$, the following holds for all $h_\tau \in \mathcal{H}_\tau$:*

$$R_{P_u}(h_\tau) \leq \hat{R}_{\mathcal{D}_v}(h_\tau) + |\hat{R}_{\mathcal{D}_v, w}(h_\tau) - \hat{R}_{\mathcal{D}_v}(h_\tau)| + 2B \cdot L \cdot \mathfrak{R}_V(\mathcal{H}_\tau) + U \cdot B \sqrt{\frac{\log(1/\delta)}{2V}}.$$

The first term $\hat{R}_{\mathcal{D}_v}$ reflects the performance on the proxy dataset, which we directly minimize when learning $\tau$. $|\hat{R}_{\mathcal{D}_v, w} - \hat{R}_{\mathcal{D}_v}|$ measures the divergence between the proxy and target distributions. This theoretically demonstrates that our DDDE method is crucial for the success of LMC. A more accurate distribution estimate leads to a smaller discrepancy term, a tighter bound, and thus a more reliable $\tau^*$ found by LMC. This links the two components of our method together. $\mathfrak{R}_V(\mathcal{H}_\tau)$ is the Rademacher complexity of the hypothesis space, which captures the richness of the classifier family. Due to page limitations, additional analysis on the discrepancy term $|\hat{R}_{\mathcal{D}_v, w} - \hat{R}_{\mathcal{D}_v}|$ and the Rademacher complexity $\mathfrak{R}_V(\mathcal{H}_\tau)$ is presented in Appendix E.

While Proposition 1 provides a probably-approximately-correct-style guarantee for our method, its form is general to many domain adaptation scenarios. To specifically highlight the unique advantages of our LMC, we delve into the convexity of the optimization objective in Appendix F, which guarantees that standard optimization methods (such as gradient descent) can efficiently and reliably converge to the unique global minimum $\tau^*$ on a limited set $\mathcal{D}_v$.

Table 1: Comparison with other LTSSL methods under the various distributions on CIFAR-10/100-LT. Top and second-best performances are bolded and underlined, respectively.

| Method | CIFAR-10-LT | | | | | CIFAR-100-LT | | | | |
| | CON | UNI | REV | MID | HT | CON | UNI | REV | MID | HT |
|---|---|---|---|---|---|---|---|---|---|---|
| Supervised | 53.27±8.94 | 61.18±2.74 | 61.73±2.55 | 53.48±8.66 | 53.64±8.84 | 47.12±1.65 | 47.09±1.66 | 46.95±1.63 | 46.83±1.58 | 47.01±1.59 |
| FixMatch | 71.93±4.05 | 71.24±2.58 | 68.30±3.14 | 70.10±4.35 | 66.50±6.84 | 55.89±1.94 | 47.40±1.66 | 55.83±2.03 | 55.65±2.01 | 56.59±1.94 |
| *Logit Adjustment-Based* | | | | | | | | | | |
| ACR | 80.89±2.92 | 82.16±1.02 | 83.49±1.15 | 79.74±4.75 | 78.42±4.45 | 58.31±1.77 | 48.98±1.45 | 59.21±1.17 | 57.66±1.64 | 58.76±1.31 |
| CPE | 80.28±3.18 | 82.59±1.13 | 84.01±0.91 | 80.80±3.05 | 79.63±3.86 | 53.12±2.19 | 46.89±1.47 | 58.61±1.12 | 54.42±2.35 | 57.15±1.56 |
| Meta-Expert | 81.33±2.53 | 83.12±1.09 | 85.03±0.54 | 80.85±3.20 | 79.87±3.44 | 54.60±2.21 | 47.85±1.55 | 58.59±1.74 | 55.09±2.08 | 57.64±1.75 |
| Sim-Pro | 81.57±2.29 | 80.86±0.83 | 74.71±1.57 | 77.69±4.22 | 76.78±2.95 | 58.30±1.60 | 48.06±1.78 | 57.36±1.89 | 57.55±1.69 | 58.36±1.57 |
| TRAS | 59.21±5.88 | 58.25±2.30 | 56.16±2.03 | 51.34±9.53 | 52.37±5.57 | 49.51±2.14 | 45.07±2.00 | 46.08±2.46 | 47.64±2.58 | 48.13±2.62 |
| *Loss Reweighting-Based* | | | | | | | | | | |
| DeBiasPL | 72.51±4.40 | 71.64±3.13 | 68.68±3.50 | 70.67±4.63 | 67.30±7.04 | 56.37±1.74 | 47.92±1.67 | 56.60±2.09 | 56.16±1.73 | 56.98±1.69 |
| SAW | 76.57±3.73 | 81.34±0.69 | 74.83±1.56 | 74.39±4.97 | 76.70±2.40 | 57.09±1.72 | 48.20±1.62 | 57.50±1.72 | 56.93±1.93 | 57.87±1.83 |
| *Resampling-Based* | | | | | | | | | | |
| ABC | 80.15±2.57 | 81.08±0.88 | 81.83±1.01 | 78.70±4.23 | 79.40±3.21 | 57.06±1.56 | 49.07±1.52 | 58.48±1.14 | 57.31±1.70 | 58.75±1.57 |
| ADSH | 73.59±3.86 | 75.16±1.47 | 68.69±1.74 | 65.45±5.70 | 73.67±2.37 | 56.48±1.83 | 48.18±1.75 | 54.35±1.87 | 54.72±2.01 | 56.27±2.08 |
| CReST | 75.16±3.84 | 82.00±1.56 | 83.60±2.69 | 79.78±4.02 | 77.71±5.93 | 56.18±1.71 | 47.50±1.63 | 57.95±1.51 | 57.47±1.37 | 56.43±1.49 |
| CReST+ | 76.10±3.16 | 81.51±1.69 | 69.62±2.84 | 78.34±4.02 | 73.05±6.35 | 56.03±1.59 | 47.62±1.71 | 57.37±1.79 | 56.81±1.79 | 56.58±1.42 |
| *Data Mixing-Based* | | | | | | | | | | |
| BEM | 75.27±3.76 | 80.07±1.53 | 77.77±1.96 | 80.29±3.45 | 73.37±4.36 | 57.58±1.67 | 49.22±1.27 | 58.30±1.67 | 57.45±1.79 | 58.05±1.51 |
| CoSSL | 80.52±2.59 | 79.42±1.64 | 74.38±2.34 | 79.32±3.44 | 72.55±4.34 | 56.90±1.33 | 49.57±1.36 | 57.23±1.80 | 57.30±1.55 | 58.06±1.68 |
| *Distribution Alignment-Based* | | | | | | | | | | |
| DARP | 74.11±3.48 | 71.59±2.72 | 68.84±3.06 | 70.80±4.37 | 67.08±6.80 | 56.73±1.66 | 47.82±1.60 | 56.39±1.91 | 55.82±1.99 | 56.93±1.84 |
| RDA | 73.14±3.42 | 77.51±1.03 | 71.80±1.38 | 64.58±5.04 | 70.84±4.03 | 56.32±1.99 | 47.64±1.81 | 56.16±1.86 | 55.53±2.31 | 57.06±1.97 |
| *Others* | | | | | | | | | | |
| DASO | 70.16±4.74 | 74.75±1.94 | 76.47±1.72 | 68.32±5.58 | 68.61±5.39 | 57.32±1.82 | 47.89±1.59 | 57.93±1.97 | 57.27±1.84 | 58.16±1.92 |
| CoLA (Ours) | **81.87±2.70** | **83.66±1.29** | **85.61±1.56** | **81.86±3.41** | **80.65±3.32** | **59.04±1.59** | **50.26±1.23** | **60.39±1.22** | **58.71±1.58** | **59.89±1.45** |

## 6 EXPERIMENT

### 6.1 DATASET

Following the settings in (Ma et al., 2024; Hou & Jia, 2025), we combine all the LTSSL methods with the classical SSL algorithm FixMatch (Sohn et al., 2020) and evaluate the performance on four widely-used datasets: CIFAR-10/100-LT (Krizhevsky, 2009), STL-10-LT (Coates et al., 2011), and Small-ImageNet-127 (SIN-127) (Fan et al., 2022). More details can be found in the Appendix G.1.

### 6.2 COMPARISON WITH OTHER METHODS

We compare CoLA with the following algorithms: **supervised learning method**, **FixMatch** (Sohn et al., 2020), and FixMatch with 6 types of SOTA LTSSL algorithms. LTSSL algorithms include 5 LA algorithms, *i.e.*, **ACR** (Wei & Gan, 2023), **CPE** (Ma et al., 2024), **Meta-Expert** (Hou & Jia, 2025), **Sim-Pro** (Du et al., 2024), and **TRAS** (Wei et al., 2024), 2 re-weighting algorithms, *i.e.*, **DeBiasPL** (Wang et al., 2022a), and **SAW** (Lai et al., 2022), 4 re-sampling algorithms, *i.e.*, **ABC** (Lee et al., 2021), **ADSH** (Guo & Li, 2022), **CReST** (Wei et al., 2021), and **CReST+** (Wei et al., 2021), 2 data mixing algorithms, *i.e.*, **BEM** (Zheng et al., 2024), and **CoSSL** (Fan et al., 2022), 4 distribution alignment algorithm, *i.e.*, **DARP** (Kim et al., 2020), **RDA** (Duan et al., 2022), **UDAL** (Lazarow et al., 2023), and **ADELLO** (Sanchez Aimar et al., 2024), and 1 other types of algorithm, *i.e.*, **DASO** (Oh et al., 2022). The results on CIFAR-10/100-LT and on STL-10-LT and SIN-127 are shown in Section 6.2.1, Section 6.2.2, and Section 6.2.3, respectively.

### 6.2.1 RESULTS ON CIFAR-10/100-LT

We evaluate CoLA against other algorithms on CIFAR-10/100-LT across 5 distributions: consistent (CON), uniform (UNI), reversed (REV), middle (MID), and head-tail (HT). For each distribution, we configure 2 or 4 distinct settings on CIFAR-10-LT, and 2 on CIFAR-100-LT. Each setting is executed 5 times with different random seeds. Due to page limitations, we report the mean and standard deviation aggregated over all runs for each distribution. For instance, the result of the consistent distribution on CIFAR-10-LT is the average of 20 individual runs (4 settings × 5 seeds). Further details on the experimental setup and results are provided in Appendix G.1 and J, respectively.

The results are presented in Table 1, from which we draw several key observations: (1) Our proposed CoLA achieves the highest accuracy across all five distributions on both the CIFAR-10-LT and CIFAR-100-LT datasets. This demonstrates its strong capability to handle diverse distributions reflective of complex, real-world scenarios; (2) The performance advantage of CoLA over other LA-based methods is more pronounced on the more challenging CIFAR-100-LT dataset, where it surpasses the runner-up by more than one percentage point in nearly all cases. We hypothesize

Table 2: Comparison with other LTSSL methods on STL-10-LT. $-$ in the $(\gamma_l, \gamma_u)$ row indicates that $\gamma_u$ is unknown. Top and second-best performances are bolded and underlined, respectively.

| $(N_1, M)$
$(\gamma_l, \gamma_u)$ | $(150, \approx 100k)$ | | $(450, \approx 100k)$ | |
|---|---|---|---|---|
| | $(10, -)$ | $(20, -)$ | $(10, -)$ | $(20, -)$ |
| Supervised | 46.62±1.54 | 41.38±0.67 | 62.79±1.42 | 55.33±1.06 |
| FixMatch | 64.96±2.85 | 55.30±0.85 | 77.10±0.57 | 69.59±0.95 |
| *Logit Adjustment-Based* | | | | |
| ACR | 71.26±0.75 | 68.13±1.51 | 79.26±0.44 | 75.90±0.68 |
| CPE | 71.18±0.62 | 67.23±2.92 | 78.71±1.10 | 76.27±0.55 |
| Meta-Expert | 71.37±0.57 | 68.04±1.34 | 79.07±1.15 | 76.46±0.89 |
| SimPro | 69.24±0.88 | 63.32±2.00 | 77.22±0.87 | 73.59±0.53 |
| TRAS | 50.29±1.50 | 42.38±1.07 | 64.78±1.00 | 55.78±0.66 |
| *Loss Reweighting-Based* | | | | |
| DeBiasPL | 69.45±0.49 | 65.90±1.06 | 74.77±0.95 | 67.83±0.69 |
| SAW | 69.26±0.73 | 65.79±0.95 | 77.76±0.60 | 74.57±1.04 |
| *Resampling-Based* | | | | |
| ABC | 70.42±0.84 | 65.74±2.33 | 78.67±0.82 | 76.03±0.74 |
| ADSH | 69.47±0.94 | 64.95±0.99 | 76.03±0.88 | 72.17±0.91 |
| CReST | 65.39±0.85 | 61.32±0.95 | 76.15±0.35 | 73.09±1.00 |
| CReST+ | 64.42±1.83 | 56.89±2.28 | 74.16±0.28 | 68.66±1.11 |
| *Data Mixing-Based* | | | | |
| BEM | 68.07±0.92 | 61.48±0.78 | 75.51±1.36 | 65.60±1.42 |
| CoSSL | 70.94±0.76 | 66.66±1.16 | 77.46±1.03 | 68.26±1.13 |
| *Distribution Alignment-Based* | | | | |
| DARP | 63.75±2.23 | 56.08±1.08 | 75.03±0.68 | 68.03±0.81 |
| RDA | 71.11±0.23 | 66.72±0.95 | 78.36±0.73 | 75.82±0.69 |
| UDAL | 69.13±0.90 | 64.26±1.42 | 77.26±1.04 | 73.74±1.15 |
| *Others* | | | | |
| DASO | 69.25±0.79 | 62.66±1.43 | 78.48±0.67 | 73.80±1.02 |
| CoLA (Ours) | **73.32±0.73** | **68.96±2.55** | **79.70±0.48** | **77.53±0.80** |

that the relative simplicity of CIFAR-10-LT may not be sufficient to fully distinguish the capabilities of highly competitive methods, which in turn highlights the efficacy of our approach on more demanding tasks; (3) While CoLA sets a new SOTA, other LA-based methods, namely Sim-Pro, Meta-Expert, and ACR, consistently secure the second-best performance across almost all the distributions, outperforming approaches from other types. This validates the general strength of the LA paradigm. Our work further advances this line of research by addressing two unique challenges for LA in the LTSSL context, thereby breaking through its performance bottleneck.

### 6.2.2 RESULTS ON STL-10-LT

We compare CoLA with other LTSSL algorithms on STL-10-LT, which is a more challenging dataset that mirrors the real-world data distribution scenarios: the unlabeled distribution is unknown, and it may contain out-of-distribution (OOD) samples The results in Table 2 represent mean accuracy and standard deviation over 5 independent runs, from which we observe that CoLA outperforms other LA-based methods in all settings. For instance, given $N_1 = 150$ with $\gamma_l = 10$ and $N_1 = 450$ with $\gamma_l = 20$, CoLA surpasses the second-best LA-based method Meta-Expert by $1.95\%$ and $1.07\%$, respectively. Additionally, we compare with ADELLO (Sanchez Aimar et al., 2024). Unlike our work, which focuses on addressing the imbalance problem in SSL, the low-confidence distillation technique in ADELLO emphasizes OOD filtering. Due to page limitations, we present the comparison results in Appendix I.

### 6.2.3 RESULTS ON SIN-127

We compare CoLA with several representative methods from other types on SIN-127. Following the settings in (Fan et al., 2022; Park et al., 2024), we conduct comparison experiments where the images are down-sampled to the sizes of $32 \times 32$ or $64 \times 64$. The results are shown in Table 3, from which we can observe that CoLA outperforms the other methods. Given that SIN-127 is a large-scale dataset, the results demonstrate that CoLA can also be applied to large-scale datasets.

## 6.3 ABLATION STUDY

To validate the effectiveness of DDDE and LMC, we conduct ablation studies on CIFAR-10/100-LT across 5 distinct distributions, where we set $(N_1, M_{k_{\max}}) = (1500, 3000)$ and $(N_1, M_{k_{\max}}) = (150, 300)$ on CIFAR-10-LT and CIFAR-100-LT, respectively. We evaluate the following variants: (1) w/o D-$\tau$: The model without DDDE, where $\tau$ is fixed to $1, 2, 4$. (2) w/o D-L: The model with only LMC, excluding DDDE. (3) w/ D-L (Ours): The full model incorporating both DDDE and LMC. For all variants without DDDE, the unlabeled class distribution is estimated using the empirical sample frequency. From the results of Table 4, we have several observations: (1) The optimal fixed $\tau$ is inconsistent across datasets, for instance, on CIFAR-10-LT, $\tau = 2$ generally performs better than $\tau = 1$, whereas the opposite trend is observed on CIFAR-100-LT. And even the best performance among w/o D-$\tau$, $\tau \in 1, 2, 4$ is still lower than w/o D-L, which indicates that using a single fixed $\tau$ causes overall adjustment to conflict with new class-level adjustments, harming final accuracy. (2) w/o D-L consistently underperforms w/ D-L, which shows that when class-level estimation is unreliable, the LMC-learned $\tau$ becomes misguided. The above observations indicate that the interaction between the two adjustments is bidirectional and also highlight the individual contribution.

Table 3: Comparison on SIN-127.

| Image Size | $32 \times 32$ | $64 \times 64$ |
|---|---|---|
| Supervised | 12.19 | 21.01 |
| FixMatch | 13.30 | 25.41 |
| ACR | 22.73 | 36.28 |
| Sim-Pro | 21.85 | 35.89 |
| DeBiasPL | 13.49 | 26.54 |
| ABC | 23.66 | 34.88 |
| ADSH | 17.28 | 23.26 |
| CReST | 12.25 | 25.59 |
| CoSSL | 13.62 | 27.02 |
| DARP | 13.51 | 26.86 |
| DASO | 13.47 | 25.97 |
| CoLA (Ours) | **24.18** | **37.49** |

Furthermore, we conduct a comparative analysis of DDDE against two alternative class distribution estimation methods: the Normalized Weighted Geometric Mean Approximation (NWGMA) (Baldi & Sadowski, 2013) and the Monte Carlo Approximation (MCA) (Park et al., 2024). Specifically, we evaluate the average $L_2$ distance between the estimated and true distributions, calculated over the final 8 training epochs on CIFAR-10/100-LT across the 5 distributions. As presented in Table 5, our proposed DDDE consistently achieves the minimum $L_2$ distance across all scenarios. This superior performance demonstrates that our method provides a more precise estimation of the unlabeled data distribution, a critical factor that significantly improves the accuracy of the pseudo-labels.

Table 4: Ablation of DDDE and LMC. $\sim$ in the $(k_{\max}, k_{\min})$ row represents an arbitrary class.

| Dataset | CIFAR-10-LT | | | | | CIFAR-100-LT | | | | |
|---|---|---|---|---|---|---|---|---|---|---|
| $(k_{\max}, k_{\min})$ | $(1, 10)$ | $(\sim, \sim)$ | $(10, 1)$ | $(5, 10)$ | $(10, 5)$ | $(1, 100)$ | $(\sim, \sim)$ | $(100, 1)$ | $(50, 100)$ | $(100, 50)$ |
| $(\gamma_l, \gamma_u)$ | $(100, 100)$ | $(100, 1)$ | $(100, 100)$ | $(100, 100)$ | $(100, 100)$ | $(10, 10)$ | $(10, 1)$ | $(10, 10)$ | $(10, 00)$ | $(10, 10)$ |
| w/o D-1 | 83.12 | 82.03 | 80.57 | 82.34 | 81.30 | 56.23 | 48.07 | 59.33 | 55.69 | 57.83 |
| w/o D-2 | 83.56 | 82.31 | 83.42 | 83.06 | 82.11 | 55.41 | 47.68 | 59.02 | 55.84 | 57.59 |
| w/o D-4 | 82.64 | 81.39 | 75.44 | 81.27 | 79.25 | 53.32 | 46.05 | 57.36 | 54.22 | 56.14 |
| w/o D-L | 84.66 | 83.38 | 84.77 | 84.50 | 83.86 | 60.16 | 50.63 | 60.79 | 59.64 | 60.18 |
| w/ D-L (Ours) | **85.04** | **84.83** | **86.84** | **85.16** | **84.42** | **60.42** | **51.28** | **61.40** | **60.10** | **61.17** |

Table 5: $L_2$ distance between the true distribution and the estimated one from different methods.

| Dataset | CIFAR-10-LT | | | | | CIFAR-100-LT | | | | |
|---|---|---|---|---|---|---|---|---|---|---|
| $(k_{\max}, k_{\min})$ | $(1, 10)$ | $(\sim, \sim)$ | $(10, 1)$ | $(5, 10)$ | $(10, 5)$ | $(1, 100)$ | $(\sim, \sim)$ | $(100, 1)$ | $(50, 100)$ | $(100, 50)$ |
| $(\gamma_l, \gamma_u)$ | $(100, 100)$ | $(100, 1)$ | $(100, 100)$ | $(100, 100)$ | $(100, 100)$ | $(10, 10)$ | $(10, 1)$ | $(10, 10)$ | $(10, 00)$ | $(10, 10)$ |
| MCA | 0.0705 | 0.0464 | 0.2564 | 0.1132 | 0.1114 | 0.0371 | 0.0431 | 0.0534 | 0.0540 | 0.0499 |
| NWGMA | 0.0616 | 0.0373 | 0.1495 | 0.1010 | 0.1097 | 0.0390 | 0.0386 | 0.0492 | 0.0505 | 0.0478 |
| DDDE (Ours) | **0.0529** | **0.0305** | **0.0891** | **0.0766** | **0.0585** | **0.0313** | **0.0353** | **0.0482** | **0.0468** | **0.0433** |

## 6.4 VISUALIZATION

In Figure 2, we visualize the accuracy of generated pseudo-labels throughout the training epochs across 5 different distributions on the CIFAR-10/100-LT. We observe a distinct trend following the application of the optimal scaling factor $\tau$, derived from our LMC (indicated by the gray dashed line). Specifically, the improvement in accuracy is most pronounced on CIFAR-10-LT under the uniform, middle, and head-tail distributions. Furthermore, a clear, yet modest, enhancement is also observed on CIFAR-100-LT for the consistent and reversed distributions. In the remaining settings,

Figure 2: Accuracy of pseudo-labels as a function of the training epoch under 5 different distributions. The blue and orange curves represent the mean accuracy over 5 independent runs on CIFAR-10-LT and CIFAR-100-LT, respectively. The vertical gray dashed line indicates the epoch at which the optimal $\tau$ derived from our meta-learning is applied for all subsequent training.

the accuracy continues to improve at a rate comparable to the phase before this application. These results demonstrate that our proposed method not only preserves the baseline pseudo-label accuracy across diverse distributions but also achieves significant improvements in specific scenarios.

# 7    CONCLUSION

In this paper, we identify two critical limitations in the existing LTSSL algorithms stemming from the design of the LA mechanism. First, their reliance on simple frequency counting overestimates the prevalence of head classes due to sample redundancy, resulting in harmful over-suppression of head classes. Second, inaccurate distribution estimation will affect the determination of overall adjustment strength, as we empirically find that the optimal overall adjustment is highly sensitive to dataset characteristics. Current research regards the overall adjustment as fixed, let alone adaptively adjusting it according to dataset characteristics. To resolve this, we introduce CoLA, which co-designs the class-wise and overall LA components. CoLA first produces a de-duplicated estimate of the unlabeled distribution and then meta-learns the optimal overall adjustment strength on a proxy set that matches the refined distribution. Supported by theoretical analysis and extensive empirical results, CoLA establishes a new SOTA in the standard long-tail setup, which offers a more principled and adaptive approach to pseudo-labeling in complex, real-world settings.

## 8 ETHICS STATEMENT

We confirm that this research strictly adheres to the ICLR Code of Ethics. Our work focuses on addressing the problem of algorithmic bias in LTSSL, which in itself contributes to improving the fairness of machine learning models. The datasets used in this study are all public computer vision benchmarks (CIFAR-10/100-LT, STL-10-LT, and SIN-127) and do not involve any sensitive or private information, nor do they involve human subjects. Our method is a general algorithmic improvement, and to the best of our knowledge, there are no potential negative social impacts or risks of misuse. Therefore, we believe that this work does not raise additional ethical concerns.

## 9 REPRODUCIBILITY STATEMENT

To ensure the reproducibility of our research, we have made significant efforts. All our experiments were conducted on four public benchmark datasets, and detailed information regarding data processing, experimental setup, model architecture, and hyperparameters is provided in Appendix G. The complete algorithmic details of our proposed CoLA method are described in Section 4. Furthermore, we have submitted our implementation code as supplementary material. We will release our code after acceptance to facilitate the community in reproducing our experimental results and building upon our work for future research.

## ACKNOWLEDGEMENTS

This research was partially supported by National Natural Science Foundation of China under grant No. T2541004, Zhejiang Key R&D Program of China under grants No. 2025C02120, No. 2024SSYS0026, Zhejiang Key Laboratory of Medical Imaging Artificial Intelligence, and the Transvascular Implantation Devices Research Institute (TIDRI) under Grant No. KY052025003.

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

## A    THE USE OF LLMS

We made light use of a large language model (Gemini 2.5) for grammar checking, spelling correction, and minor language polishing. The model was **not** involved in research ideation, experimental design, analysis, or interpretation. All scientific contributions and conclusions are entirely the responsibility of the authors.

## B    SUPPLEMENTARY RELATED WORK

This section supplements Section 2, including five other types of LTSSL algorithms:

- **Loss re-weighting** (Ross & Dollár, 2017) increases (decreases) the weight of tail (head) class samples in the training loss, ensuring the model learns better decision boundaries. SAW (Lai et al., 2022) smoothly re-weights the training loss based on the estimated learning difficulty. DebiasPL (Wang et al., 2022a) designs the adaptive marginal loss to mitigate the pseudo-label bias.

- **Re-sampling** (Chawla et al., 2002; He & Garcia, 2009) addresses the bias by constructing balanced training batches to mitigate head-class dominance. ABC (Lee et al., 2021) achieves this through class-aware Bernoulli sampling, a paradigm extended by RECD (Park et al., 2024), which incorporates a Monte Carlo estimation of the unlabeled data distribution for more precise balancing. Other strategies focus on re-balancing specific data pools. CReST (Wei et al., 2021) prioritizes the labeled data distribution, augmenting it with high-confidence pseudo-labeled samples. Based on this, CReST+ (Wei et al., 2021) adopts a progressive distribution alignment to adaptively adjust the rebalancing strength. In contrast, ADSH (Guo & Li, 2022) concentrates on the unlabeled set, generating class-balanced pseudo-labels via an adaptive thresholding mechanism.

- **Data mixing** (Yun et al., 2019; Zhang et al., 2018) performs mixing at element-wise or region-wise levels to create diverse training samples, which alleviates the problem of data imbalance. BEM (Zheng et al., 2024) presents a balanced and entropy-based mixing strategy to re-balance the class-wise uncertainty of training samples. CoSSL (Fan et al., 2022) proposes a tail-class feature enhancement that expands the dataset by blending unlabeled data features with labeled data features while preserving the label of the labeled samples.

- **Distribution Alignment** (Berthelot et al., 2020) reduces the confirmation bias via regularization added to the pseudo-label during inference. DARP (Kim et al., 2020) models the pseudo-label distribution alignment problem as a convex optimization problem and iteratively soft-corrects pseudo-labels to make their distribution approach the target class prior. RDA (Duan et al., 2022) introduces an auxiliary classifier to mutually align the pseudo-label distribution with the complementary label distribution, thereby addressing the challenge of mismatched class distributions between labeled and unlabeled data without requiring any prior distribution information. UDAL (Lazarow et al., 2023) unifies distribution alignment and LA, offering a loss-level perspective. ADELLO (Sanchez Aimar et al., 2024) collaboratively employs dynamic distribution alignment and low-confidence sample distillation to alleviate the issue of label distribution shift.

- **Others.** Beyond these core directions, innovative frameworks address label bias through alternative lenses. DASO (Oh et al., 2022) obtains unbiased pseudo-labels by fusing predictions from similarity-based and linear classifiers. CISSL (Kim et al., 2025) computes independent contrastive losses for labeled and unlabeled samples, respectively, aiming to leverage the ground-truth information of labeled samples to regularize feature representations and mitigate confirmation bias. SoftMatch (Chen et al., 2023), while designed for standard SSL, addresses the quantity-quality trade-off relevant to pseudo-labeling in LTSSL.

## C    TECHNICAL LEMMAS

**Lemma 1** (Theorem 3.3 (Mohri et al., 2018)). *Let $\mathcal{G}$ be a family of functions mapping from an arbitrary input space $\mathcal{Z}$ to $[0, 1]$. Then for any $\delta > 0$, with probability at least $1 - \delta$ over the draw of an i.i.d. sample $S$ of size $m$, each of the following holds for all $g \in \mathcal{G}$:*

$$\mathbb{E}[g(z)] \leq \frac{1}{m} \sum_{i=1}^{m} g(z_i) + 2\mathfrak{R}_m(\mathcal{G}) + \sqrt{\frac{\log \frac{1}{\delta}}{2m}}.$$

**Corollary 1.** *Let $\mathcal{G}$ be a family of functions mapping from $\mathcal{Z}$ to $[a, b]$. Then for any $\delta > 0$, with probability at least $1 - \delta$ over the draw of an i.i.d. sample $S$ of size $m$, each of the following holds for all $g \in \mathcal{G}$:*

$$\mathbb{E}[g(z)] \leq \frac{1}{m} \sum_{i=1}^{m} g(z_i) + 2\mathfrak{R}_m(\mathcal{G}) + (b-a)\sqrt{\frac{\log \frac{1}{\delta}}{2m}}.$$

*Proof.* We could straightly follow the proof for Lemma 1 given by (Mohri et al., 2018) to derive Corollary 1, without any other technique. The detailed proof is omitted. □

**Lemma 2** (Talagrand's lemma (Mohri et al., 2018)). *Let $\Phi_1, ..., \Phi_m$ be $l$-Lipschitz functions from $\mathbb{R}$ to $\mathbb{R}$ and $\sigma_1, ..., \sigma_m$ be Rademacher random variables. Then, for any hypothesis set $\mathcal{H}$ of real-valued functions, the following inequality holds:*

$$\frac{1}{m} \mathbb{E}_\sigma \left[ \sup_{h \in \mathcal{H}} \sum_{i=1}^{m} \sigma_i (\Phi_i \circ h)(x_i) \right] \leq \frac{l}{m} \mathbb{E}_\sigma \left[ \sup_{h \in \mathcal{H}} \sum_{i=1}^{m} \sigma_i h(x_i) \right] = l\,\widehat{\mathfrak{R}}_S(\mathcal{H}).$$

*In particular, if $\Phi_i = \Phi$ for all $i \in [m]$, then the following holds:*

$$\widehat{\mathfrak{R}}_S(\Phi \circ \mathcal{H}) \leq l\,\widehat{\mathfrak{R}}_S(\mathcal{H}).$$

## D  PROOF OF PROPOSITION 1

Our proof relies on standard results from statistical learning theory, particularly generalization bounds for importance-weighted risks (Shimodaira, 2000).

*Proof of Proposition 1.* First, we express the target risk $R_{P_u}(h_\tau)$ as an importance-weighted expectation over the proxy distribution $P_v$.

$$\begin{aligned}
R_{P_u}(h_\tau) &= \mathbb{E}_{(\boldsymbol{X}_u, Y_u) \sim P_u}[\ell(h_\tau(\boldsymbol{X}_u), Y_u)] \\
&= \int_{\mathcal{X} \times \mathcal{Y}} \ell(h_\tau(\boldsymbol{x}), y) p_u(\boldsymbol{x}, y) \mathrm{d}\boldsymbol{x}\mathrm{d}y \\
&= \int_{\mathcal{X} \times \mathcal{Y}} \ell(h_\tau(\boldsymbol{x}), y) p_u(\boldsymbol{x}|y) p_u(y) \mathrm{d}\boldsymbol{x}\mathrm{d}y \\
&= \int_{\mathcal{X} \times \mathcal{Y}} \ell(h_\tau(\boldsymbol{x}), y) p_u(\boldsymbol{x}|y) p_u(y) \frac{p_v(y)}{p_v(y)} \mathrm{d}\boldsymbol{x}\mathrm{d}y \\
&= \int_{\mathcal{X} \times \mathcal{Y}} \ell(h_\tau(\boldsymbol{x}), y) p_v(\boldsymbol{x}, y) \frac{p_u(y)}{p_v(y)} \mathrm{d}\boldsymbol{x}\mathrm{d}y \\
&= \int_{\mathcal{X} \times \mathcal{Y}} w(y) \ell(h_\tau(\boldsymbol{x}), y) p_v(\boldsymbol{x}, y) \mathrm{d}\boldsymbol{x}\mathrm{d}y \\
&= \mathbb{E}_{(\boldsymbol{X}_v, Y_v) \sim P_v}[w(y) \ell(h_\tau(\boldsymbol{X}_v), Y_v)],
\end{aligned}$$

where $p_u(\boldsymbol{x}, y)$, $p_u(\boldsymbol{x}|y)$, and $p_u(y)$ denote the probability density function (PDF) of $P_u$, class-conditional PDF of $P_{\boldsymbol{X}_u|Y_u}(\boldsymbol{x}|y)$, and class marginal PDF of $P_{Y_u}(y)$, respectively. While $p_v(\boldsymbol{x}, y)$, $p_v(y)$ denote the PDF of $P_v$, and class marginal PDF of $P_{Y_v}(y)$, respectively.

Let us define a family of weighted loss functions $\mathcal{G} = \{g : (\boldsymbol{x}, y) \mapsto w(y) \ell(h_\tau(\boldsymbol{x}), y) \mid h_\tau \in \mathcal{H}_\tau\}$. By Assumption 1, $\ell(h_\tau(\boldsymbol{x}), y) \in [0, U]$, and by Assumption 4, $w(y) \in [0, B]$. Therefore, any function $g \in \mathcal{G}$ maps to the interval $[0, U \cdot B]$. We can apply Corollary 1 to the family $\mathcal{G}$ and the sample $\mathcal{D}_v$. For any $\delta \in (0, 1)$, with probability at least $1 - \delta$, we have for all $g \in \mathcal{G}$:

$$\mathbb{E}[g(\boldsymbol{X}_v, Y_v)] \leq \frac{1}{V} \sum_{i=1}^{V} g(\boldsymbol{x}_i^v, y_i^v) + 2\mathfrak{R}_V(\mathcal{G}) + U \cdot B \sqrt{\frac{\log(1/\delta)}{2V}}.$$

Substituting the definitions of $g$, we get:

$$R_{P_u}(h_\tau) \leq \hat{R}_{\mathcal{D}_v,w}(h_\tau) + 2\mathfrak{R}_V(\mathcal{G}) + U \cdot B\sqrt{\frac{\log(1/\delta)}{2V}}. \tag{2}$$

Next, we bound the Rademacher complexity $\mathfrak{R}_V(\mathcal{G})$. Define $\mathcal{H}_\ell$ as the family of loss functions associated to $\mathcal{H}_\tau$ mapping from $\mathcal{X} \times \mathcal{Y}$ to $\mathbb{R}$: $\mathcal{H}_\ell = \{h_\ell : (\boldsymbol{x}, y) \mapsto \ell(h_\tau(\boldsymbol{x}), y) \mid h_\tau \in \mathcal{H}_\tau\} = \ell \circ \mathcal{H}_\tau$. Define a mapping $\Phi_y : \mathbb{R} \to \mathbb{R}$ such that $\Phi_y(a) = w(y) \cdot a$. The function family $\mathcal{G}$ can be viewed as the composition of each function $\ell(h_\tau(\boldsymbol{x}), y)$ in $\mathcal{H}_\ell$ with the $y$-dependent function $\Phi_y$, denoted by $\mathcal{G} = \Phi_y \circ \mathcal{H}_\ell$. Since $\Phi_y$ is linear, and its derivative is $w(y) \leq B$ (Assumption 4), $\Phi_y$ is $B$-Lipschitz continuous. Now, we apply Lemma 2 to $\mathcal{G} = \Phi_y \circ \mathcal{H}_\ell$:

$$\mathfrak{R}_V(\mathcal{G}) = \mathfrak{R}_V(\Phi_y \circ \mathcal{H}_\ell) \leq B \cdot \mathfrak{R}_V(\mathcal{H}_\ell). \tag{3}$$

Although in the canonical form of Lemma 2 the term $\Phi$ does not depend on the samples, it is obvious that the result continues to hold when expressed in this multiplicative form that depends on a subset of the samples (here, $y$). Since $\ell$ is $L$-Lipschitz (Assumption 1), we apply Lemma 2 to $\mathcal{H}_l = \ell \circ \mathcal{H}_\tau$:

$$\mathfrak{R}_V(\mathcal{H}_\ell) = \mathfrak{R}_V(\ell \circ \mathcal{H}_\tau) \leq L \cdot \mathfrak{R}_V(\mathcal{H}_\tau). \tag{4}$$

Combining (3) and (4) gives:

$$\mathfrak{R}_V(\mathcal{G}) \leq B \cdot L \cdot \mathfrak{R}_V(\mathcal{H}_\tau). \tag{5}$$

Substituting (5) back into (2), we get:

$$R_{P_u}(h_\tau) \leq \hat{R}_{\mathcal{D}_v,w}(h_\tau) + 2B \cdot L \cdot \mathfrak{R}_V(\mathcal{H}_\tau) + U \cdot B\sqrt{\frac{\log(1/\delta)}{2V}}.$$

Finally, we use the triangle inequality by adding and subtracting the empirical risk:

$$R_{P_u}(h_\tau) \leq \hat{R}_{\mathcal{D}_v,w}(h_\tau) + \left(\hat{R}_{\mathcal{D}_v}(h_\tau) - \hat{R}_{\mathcal{D}_v}(h_\tau)\right) + 2B \cdot L \cdot \mathfrak{R}_V(\mathcal{H}_\tau) + U \cdot B\sqrt{\frac{\log(1/\delta)}{2V}}$$

$$\leq \hat{R}_{\mathcal{D}_v}(h_\tau) + \left|\hat{R}_{\mathcal{D}_v,w}(h_\tau) - \hat{R}_{\mathcal{D}_v}(h_\tau)\right| + 2B \cdot L \cdot \mathfrak{R}_V(\mathcal{H}_\tau) + U \cdot B\sqrt{\frac{\log(1/\delta)}{2V}}.$$

This completes the proof of Proposition 1.

$\square$

## E    DETAILED ANALYSIS OF PROPOSITION 1

**Discrepancy term $|\hat{R}_{\mathcal{D}_v,w} - \hat{R}_{\mathcal{D}_v}|$.**    The discrepancy term $\left|\hat{R}_{\mathcal{D}_v,w}(h_\tau) - \hat{R}_{\mathcal{D}_v}(h_\tau)\right|$ can be controlled via the class-proportion error:

$$\left|\hat{R}_{\mathcal{D}_v,w}(h_\tau) - \hat{R}_{\mathcal{D}_v}(h_\tau)\right| = \left|\frac{1}{V}\sum_{i=1}^{V}\left(w(y_i^v) - 1\right)\ell(h_\tau(\boldsymbol{x}_i^v), y_i^v)\right| \leq \frac{U}{V}\sum_{i=1}^{V}\left|w(y_i^v) - 1\right|.$$

Because $w(y) = \frac{P_{Y_u}(y)}{P_{Y_v}(y)}$ and $P_{Y_v}(y) = \hat{P}_{Y_u}(y)$ by Assumption 2, taking expectations over $\mathcal{D}_v$ gives

$$\mathbb{E}_{\mathcal{D}_v}\left[\frac{U}{V}\sum_{i=1}^{V}\left|w(y_i^v) - 1\right|\right] = U\sum_{y \in \mathcal{Y}}\left|P_{Y_u}(y) - \hat{P}_{Y_u}(y)\right| = U\left\|P_{Y_u} - \hat{P}_{Y_u}\right\|_1.$$

Moreover, since $0 \leq w(y) \leq B$ (Assumption 4), the term $|w(y_i^v) - 1|$ is bounded by $\max(1, B-1)$. By applying Hoeffding's inequality to the empirical mean $\frac{1}{V}\sum_i^V U \cdot |w(y_i^v) - 1|$, we have that with probability at least $1 - \delta$:

$$\left|\hat{R}_{\mathcal{D}_v,w}(h_\tau) - \hat{R}_{\mathcal{D}_v}(h_\tau)\right| \leq U\left(\left\|P_{Y_u} - \hat{P}_{Y_u}\right\|_1 + \max(1, B-1)\sqrt{\frac{\log(1/\delta)}{2V}}\right).$$

Thus, when our estimator $\hat{P}_{Y_u}$ is accurate (small $L_1$ error), the discrepancy term contracts at the same rate, formally tightening the overall bound and validating the methodology.

**Rademacher complexity** $\mathfrak{R}_V(\mathcal{H}_\tau)$. To simplify the proof, denote the backbone logits (kept fixed when optimizing $\tau$) by $s(\boldsymbol{x}) \in \mathbb{R}^K$ and write $s_y(\boldsymbol{x})$ for the $y$-th component. Our LMC applies a linear correction $-\tau\,\boldsymbol{p}$ with $\boldsymbol{p} = (\hat{P}_{Y_u}(1), \ldots, \hat{P}_{Y_u}(K))$. Hence

$$h_\tau(\boldsymbol{x})_y = s_y(\boldsymbol{x}) - \tau\,p_y, \qquad p_y := \hat{P}_{Y_u}(y) \geq 0.$$

The relevant real-valued hypothesis class is therefore

$$\mathcal{H}_\tau = \Big\{ (\boldsymbol{x}, y) \mapsto s_y(\boldsymbol{x}) - \tau\,p_y \,\Big|\, \tau \in \mathcal{T} \Big\},$$

where $\mathcal{T} = [0, \tau_{\max}]$ is the search interval used in practice (without a bounded interval, the complexity would be unbounded).

For $\mathcal{D}_v$, the empirical complexity satisfies

$$\widehat{\mathfrak{R}}_{\mathcal{D}_v}(\mathcal{H}_\tau) = \mathbb{E}_{\boldsymbol{\sigma}}\left[ \sup_{\tau \in \mathcal{T}} \frac{1}{V} \sum_{i=1}^{V} \sigma_i\big(s_{y_i^v}(\boldsymbol{x}_i^v) - \tau\,p_{y_i^v}\big) \right],$$

with $\sigma_i \in \{-1, +1\}$. Because the logits are constant with respect to $\tau$, this splits into

$$\widehat{\mathfrak{R}}_{\mathcal{D}_v}(\mathcal{H}_\tau) = \frac{1}{V}\mathbb{E}_{\boldsymbol{\sigma}}\left[ \sum_{i=1}^{V} \sigma_i s_{y_i^v}(\boldsymbol{x}_i^v) \right] + \frac{1}{V}\mathbb{E}_{\boldsymbol{\sigma}}\left[ \sup_{\tau \in [0,\tau_{\max}]} \left( -\tau \sum_{i=1}^{V} \sigma_i p_{y_i^v} \right) \right]$$

$$= \frac{1}{V}\mathbb{E}_{\boldsymbol{\sigma}}\left[ \sup_{\tau \in [0,\tau_{\max}]} \left( -\tau \sum_{i=1}^{V} \sigma_i p_{y_i^v} \right) \right],$$

because $\mathbb{E}[\sigma_i] = 0$. Let $S_p := \sum_{i=1}^{V} \sigma_i p_{y_i^v}$. If $S_p \geq 0$ the supremum is achieved at $\tau = 0$; otherwise it is achieved at $\tau_{\max}$, yielding $\tau_{\max}|S_p|$. Consequently

$$\widehat{\mathfrak{R}}_{\mathcal{D}_v}(\mathcal{H}_\tau) \leq \frac{\tau_{\max}}{V}\,\mathbb{E}_{\boldsymbol{\sigma}}[|S_p|]. \tag{6}$$

Let $n_y$ be the count of class $y$ inside $\mathcal{D}_v$. Then

$$S_p = \sum_{y \in \mathcal{Y}} p_y \left( \sum_{i:y_i^v = y} \sigma_i \right) \quad \text{and} \quad \mathbb{E}_{\boldsymbol{\sigma}}\big[S_p^2\big] = \sum_{y \in \mathcal{Y}} n_y p_y^2.$$

By Khintchine's inequality,

$$\mathbb{E}_{\boldsymbol{\sigma}}[|S_p|] \leq \sqrt{\mathbb{E}_{\boldsymbol{\sigma}}\big[S_p^2\big]} = \sqrt{\sum_{y \in \mathcal{Y}} n_y p_y^2}. \tag{7}$$

Combining (6) and (7) yields the finite-sample bound

$$\widehat{\mathfrak{R}}_{\mathcal{D}_v}(\mathcal{H}_\tau) \leq \frac{\tau_{\max}}{V} \sqrt{\sum_{y \in \mathcal{Y}} n_y p_y^2}.$$

Taking expectation over draws of $\mathcal{D}_v$ and applying Jensen's inequality to $\mathbb{E}\big[\sqrt{\sum_{y \in \mathcal{Y}} n_y p_y^2}\big]$ yields

$$\mathfrak{R}_V(\mathcal{H}_\tau) \leq \frac{\tau_{\max}}{\sqrt{V}} \sqrt{\sum_{y \in \mathcal{Y}} p_y^3}. \tag{8}$$

Using (8) inside Proposition 1 leads to

$$R_{P_u}(h_\tau) \leq \hat{R}_{\mathcal{D}_v}(h_\tau) + \big|\hat{R}_{\mathcal{D}_v,w}(h_\tau) - \hat{R}_{\mathcal{D}_v}(h_\tau)\big| + \frac{2BL\tau_{\max}}{\sqrt{V}} \sqrt{\sum_{y \in \mathcal{Y}} p_y^3} + UB\sqrt{\frac{\log(1/\delta)}{2V}}.$$

From the above inequality, we can draw the following observations: (1) The third term scales as $1/\sqrt{V}$ and depends only on the search radius $\tau_{\max}$ and the geometry of the target class prior, not on the underlying deep backbone. (2) The quantity $\sum_{y \in \mathcal{Y}} p_y^3$ (a third moment of the class prior) shrinks when the distribution is closer to uniform and remains bounded even under heavy imbalance. (3) Without $\tau \in [0, \tau_{\max}]$, $\mathfrak{R}_V(\mathcal{H}_\tau)$ would diverge. In practice, the meta-search already constrains $\tau$, which simultaneously guarantees a finite complexity term. Consequently, the hypothesis class induced by meta-calibrating a single scalar admits the explicit control (8).

## F    CONVEXITY OF THE OPTIMIZATION OBJECTIVE

A significant and unique advantage of our LMC method lies in the optimization landscape for the parameter $\tau$. Recall that we find the optimal $\tau^*$ by minimizing the CE loss on the proxy set $\mathcal{D}_v$:

$$\tau^* = \arg\min_{\tau} \mathcal{L}(\tau) = \arg\min_{\tau} \frac{1}{V} \sum_{i=1}^{V} \mathcal{L}_{\text{CE}}\Big(y_i^v, \sigma(z(\alpha(\boldsymbol{x}_i^v)) - \tau \cdot \boldsymbol{p})\Big).$$

For this optimization, the logits $z(\alpha(\boldsymbol{x}_i^v))$ and the estimated class distribution vector $\boldsymbol{p}$ are treated as fixed constants. The objective function $\mathcal{L}(\tau)$ is therefore a function of a single scalar variable $\tau$. We can formally state that $\mathcal{L}(\tau)$ is a convex function with respect to $\tau$. This convexity can be established by recognizing that the CE loss is equivalent to the negative log-likelihood of the $\mathrm{softmax}$ function. The log-sum-exp function is a well-known convex function. Specifically, our objective function for a single sample can be expressed as:

$$\mathcal{L}_i(\tau) = -\big(z(\alpha(\boldsymbol{x}_i^v))_{y_i^v} - \tau \cdot \boldsymbol{p}_{y_i^v}\big) + \log \sum_{k=1}^{K} \exp\big(z(\alpha(\boldsymbol{x}_i^v))_k - \tau \cdot \boldsymbol{p}_k\big),$$

where the subscripts of $z(\alpha(\boldsymbol{x}_i^v))$ and $\boldsymbol{p}$ mean to get the element value corresponding to the respective class indices. The first term is linear in $\tau$, and the second term is a log-sum-exp function of an affine transformation of $\tau$. Since the composition of a convex function with an affine mapping is convex, and the sum of convex functions remains convex, the total objective $\mathcal{L}(\tau) = \frac{1}{V}\sum_i^V \mathcal{L}_i(\tau)$ is convex. The convexity of $\mathcal{L}(\tau)$ is a powerful property with profound practical implications:

- **Guaranteed Global Optimum:** Unlike the highly non-convex and complex loss landscape encountered when fine-tuning a deep neural network, our objective for $\tau$ has no spurious local minima. This guarantees that standard optimization methods (like gradient descent) can efficiently and reliably converge to the unique global minimum $\tau^*$.

- **Efficiency and Robustness:** The optimization is performed over a single scalar, making it computationally trivial and insensitive to initialization.

## G    DETAILS OF DATASETS AND IMPLEMENTATIONS

### G.1    DATASETS

We conduct experiments on CIFAR-10/100-LT, STL-10-LT, and SIN-127. The detailed dataset splits are as follows:

**CIFAR-10-LT** with 10 classes contains $50,000$ images for training and $10,000$ for testing. We conduct experiments on CIFAR-10-LT under 5 distributions, including consistent, uniform, reversed, middle, and head-tail distributions. Each one contains 2 or 4 settings, as shown in Table 6.

**CIFAR-100-LT** with 100 classes contains $50,000$ images for training and $10,000$ for testing. We conduct experiments on CIFAR-100-LT under 5 distributions, including consistent, uniform, reversed, middle, and head-tail distributions. Each one contains 2 settings, as shown in Table 7.

**STL-10-LT** with 10 classes contains $5,000$ images for training and $8,000$ images for testing, and $100,000$ unlabeled images as extra training data. The detailed settings are shown in Table 8.

**SIN-127** is a down-sampled version of ImageNet-127 (Huh et al., 2016), which is created by consolidating the $1,000$ classes from ImageNet (Russakovsky et al., 2015) into 127 classes. SIN-127 has $1,281,167$ training samples with an imbalance ratio of approximately 286. Following Fan et al. (2022), we randomly select $10\%$ of the samples in each class as labeled samples. The samples are down-sampled to sizes of $32 \times 32$ and $64 \times 64$. The class distribution of the test set of SIN-127 is also imbalanced. The detailed settings are shown in Table 9.

### G.2    IMPLEMENTATIONS

We provide the pseudocode for CoLA in Algorithm 1, where we use two LA strategies, including the loss modification technique for LA in line 18, and post-hoc LA elsewhere, while CoLA improves

Table 6: Detailed settings of CIFAR-10-LT.

| No. | Distributions | $(k_{\max}, k_{\min})$ | $(N_1, M_{k_{\max}})$ | $(\gamma_l, \gamma_u)$ |
|---|---|---|---|---|
| 1 | | | $(1500, 3000)$ | $(100, 100)$ |
| 2 | Consistent | $(1, 10)$ | | $(150, 150)$ |
| 3 | | | $(500, 4000)$ | $(100, 100)$ |
| 4 | | | | $(150, 150)$ |
| 5 | Uniform | (Arbitrary, Arbitrary) | $(1500, 300)$ | $(100, 1)$ |
| 6 | | | | $(150, 1)$ |
| 7 | Reversed | $(10, 1)$ | $(1500, 3000)$ | $(100, 100)$ |
| 8 | | | | $(150, 150)$ |
| 9 | | | $(1500, 3000)$ | $(100, 100)$ |
| 10 | Middle | $(5, 10)$ | | $(150, 150)$ |
| 11 | | | $(500, 4000)$ | $(100, 100)$ |
| 12 | | | | $(150, 150)$ |
| 13 | | | $(1500, 3000)$ | $(100, 100)$ |
| 14 | Head-Tail | $(10, 5)$ | | $(150, 150)$ |
| 15 | | | $(500, 4000)$ | $(100, 100)$ |
| 16 | | | | $(150, 150)$ |

Table 7: Detailed settings of CIFAR-100-LT.

| No. | Distributions | $(k_{\max}, k_{\min})$ | $(N_1, M_{k_{\max}})$ | $(\gamma_l, \gamma_u)$ |
|---|---|---|---|---|
| 1 | Consistent | $(1, 100)$ | $(150, 300)$ | $(10, 10)$ |
| 2 | | | | $(15, 15)$ |
| 3 | Uniform | (Arbitrary, Arbitrary) | $(150, 30)$ | $(10, 1)$ |
| 4 | | | | $(15, 1)$ |
| 5 | Reversed | $(100, 1)$ | | $(10, 10)$ |
| 6 | | | | $(15, 15)$ |
| 7 | Middle | $(50, 100)$ | $(150, 300)$ | $(10, 10)$ |
| 8 | | | | $(15, 15)$ |
| 9 | Head-Tail | $(100, 50)$ | | $(10, 10)$ |
| 10 | | | | $(15, 15)$ |

Table 8: Detailed settings of STL-10-LT. $-$ denotes *unknown*.

| No. | $(N_1, M)$ | $(\gamma_l, \gamma_u)$ |
|---|---|---|
| 1 | $(150, 100k)$ | $(10, -)$ |
| 2 | | $(20, -)$ |
| 3 | $(450, 100k)$ | $(10, -)$ |
| 4 | | $(20, -)$ |

Table 9: Detailed settings of SIN-127.

| No. | Image Size | $(N_1, M)$ | $(\gamma_l, \gamma_u)$ |
|---|---|---|---|
| 1 | $32 \times 32$ | $(27760, 1281167)$ | $(\approx 286, \approx 286)$ |
| 2 | $64 \times 64$ | | |

upon post-hoc LA. We combine all the LTSSL methods with the classical FixMatch (Sohn et al., 2020), which are trained and evaluated using the codebase Unified SSL Benchmark (USB) (Wang et al., 2022b) on $8\times$NVIDIA RTX 4090 with 24 GB memory. For training, we use ResNet-50 (He et al., 2016) on the SIN-127 experiment and Wide ResNet-28-2 (Zagoruyko & Komodakis, 2016) on other datasets. The parameter settings are as follows: $\rho = 0.95$, $\beta = 0.999$, $\tau_w = 2$. The optimizer for all experiments is SGD (Sutskever et al., 2013) with an initial learning rate of 0.03. The learning rate decay and momentum are 0.9 and 0.0005, respectively. The training epoch $E$ is set to 200 for experiments on SIN-127, and 256 for experiments on other datasets. For our proposed CoLA, the hyperparameter $\tau$ is set according to ACR (Wei & Gan, 2023) during the initial stage of model training. For the final 16 epochs, its value is then determined using the LMC method. Therefore, we set $E_3 = E - 16$, $E_2 = E_3 - 32$, $E_{\text{DDDE}} = 5$. For evaluation, we calculate the balanced accuracy on the test set and average our results over five independent runs, *i.e.*, seed $= 0, 1, 2, 3, 4$.

---

**Algorithm 1** Co-Calibrated Logit Adjustment (CoLA)

---

**Require:** labeled set $\mathcal{D}_l$, unlabeled set $\mathcal{D}_u$, confidence $\rho$, epochs $E$, stage cutoffs ($E_{\text{DDDE}}, E_2, E_3$), EMA rate $\beta$, warm-up $\tau_w$, standard branch $f$, balanced branch $\tilde{f}$.

1: Initialize: $\hat{P}_{Y_u} \leftarrow \mathbf{1}/K$, $\tau_{\min} \leftarrow 0$, $\tau \leftarrow \tau_w$, $\tilde{\tau} \leftarrow \tau_w$.
2: **for** $e = 1$ to $E$ **do**
3:     **if** $e < E_{\text{DDDE}}$ **then**                                  ▷ Warm-up
4:         $\boldsymbol{a}_{\text{std}} \leftarrow \tau \cdot \log P_{Y_l}$.
5:     **else if** $E_{\text{DDDE}} \leq e < E_3$ **then**        ▷ KL-guided interpolation (Wei & Gan, 2023)
6:         Compute symmetric KL scores between $\hat{P}_{Y_u}$ and 3 distributions (consistent, uniform and reversed).
7:         $\tau \leftarrow \tau_{\min} + (\tau_w - \tau_{\min}) \cdot \text{softmax}(\text{KL})_1$.
8:         $\boldsymbol{a}_{\text{std}} \leftarrow \tau \cdot \log P_{Y_l}$.
9:     **else**                                        ▷ Meta-learned logit adjustment
10:         $\boldsymbol{a}_{\text{std}} \leftarrow \tilde{\tau} \cdot \hat{P}_{Y_u}$.
11:     **end if**
12:     **for** mini-batch $(\mathcal{B}_l, \mathcal{B}_u)$ **do**
13:         Run $f, \tilde{f}$ on each labeled sample $\boldsymbol{x}_i^l \in \mathcal{B}_l$ to get logits $z(\alpha(\boldsymbol{x}_i^l)), \tilde{z}(\alpha(\boldsymbol{x}_i^l))$.
14:         $\mathcal{L}_l \leftarrow \text{CE}(y_i, \sigma(z(\alpha(\boldsymbol{x}_i^l))))$,   $\mathcal{L}_l^b \leftarrow \text{CE}(y_i, \tilde{z}(\alpha(\boldsymbol{x}_i^l)) + \tau_w \cdot \log P_{Y_l})$.
15:         Run $f, \tilde{f}$ on unlabeled sample $\boldsymbol{x}_j^u \in \mathcal{B}_u$ with weak and strong views to get logits $z(\alpha(\boldsymbol{x}_j^u)), z(\mathcal{A}(\boldsymbol{x}_j^u)), \tilde{z}(\alpha(\boldsymbol{x}_j^u)), \tilde{z}(\mathcal{A}(\boldsymbol{x}_j^u))$ and unlabeled features.
16:         $\mathbf{q}_j \leftarrow \sigma(z(\alpha(\boldsymbol{x}_j^u)) - \boldsymbol{a}_{\text{std}})$,   $\hat{y}_j \leftarrow \arg\max \mathbf{q}_j$,   $m_j \leftarrow \mathbb{I}(\max \mathbf{q}_j \geq \rho)$.
17:         $\tilde{\mathbf{q}}_j \leftarrow \sigma(z(\alpha(\boldsymbol{x}_j^u)) - \tau_w \cdot \log P_{Y_l})$, $\hat{y}_j^b \leftarrow \arg\max \tilde{\mathbf{q}}_j$, $\tilde{m}_j \leftarrow \mathbb{I}(\max \sigma(\tilde{z}(\alpha(\boldsymbol{x}_j^u))) \geq \rho)$.
18:         **if** $E_{\text{DDDE}} \leq e$ **then**                             ▷ DDDE
19:            Within each class $y$, *i.e.*, $\hat{y}_j^b = y$, gather unlabeled features $\boldsymbol{z}_j^y$ with $\tilde{m}_j = 1$, then forming a feature matrix $\boldsymbol{Z}_y$ and compute $\text{erank}(\boldsymbol{Z}_y)$.
20:         **end if**
21:         **if** $E_2 \leq e < E_3$ **then**                           ▷ LMC
22:            Sample proxy set $\mathcal{D}_v$ by Bernoulli mask $p_i \propto \hat{P}_{Y_u}(y_i^l)/P_{Y_l}(y_i^l)$.
23:            $\mathcal{L}_\tau \leftarrow \text{CE}(y_i^v, \sigma(z(\alpha(\boldsymbol{x}_i^v))) - \tilde{\tau} \cdot \hat{P}_{Y_u})$, update $\tilde{\tau}$ via SGD on $\mathcal{L}_\tau$.
24:         **end if**
25:         $\hat{P}_{Y_u} \leftarrow \beta \hat{P}_{Y_u} + (1 - \beta) \frac{\text{erank}(\boldsymbol{Z}_y)}{\sum_{k \in \mathcal{Y}} \text{erank}(\boldsymbol{Z}_k)}$.
26:         $\mathcal{L}_u \leftarrow \text{CE}(\hat{y}_j, z(\mathcal{A}(\boldsymbol{x}_j^u)))$ with mask $m_j \vee \mathbb{I}(\max \sigma(z(\alpha(\boldsymbol{x}_j^u))) \geq \rho)$.
27:         $\mathcal{L}_u^b \leftarrow \text{CE}(\hat{y}_j^b, \tilde{z}(\mathcal{A}(\boldsymbol{x}_j^u)))$ with mask $\tilde{m}_j \vee \mathbb{I}(\max \tilde{\mathbf{q}}_j \geq \rho)$.
28:         Update parameters on $\mathcal{L} = \mathcal{L}_l + \mathcal{L}_l^b + \mathcal{L}_u + \mathcal{L}_u^b$.
29:     **end for**
30: **end for**
31: **return** balanced branch $\tilde{f}$.

---

## H   Time Complexity Analysis

**DDDE Time Complexity.** Let $\mathcal{M} = \sum_y m_y$ be the total number of high-confidence samples per class in each batch, not exceeding the unlabeled batch size $|\mathcal{B}_u|$. In each training batch, DDDE

first stacks the high-confidence unlabeled sample representations of each class $y$ into a matrix $\boldsymbol{Z}_y \in \mathbb{R}^{d \times m_y}$, then performs Singular Value Decomposition (SVD) and computes entropy and $\mathrm{erank}$: (1) Constructing the representation matrix by duplicating each high-confidence sample once costs $O(d\mathcal{M})$; (2) The cost of performing full SVD on $\boldsymbol{Z}_y$ is $O(\min(dm_y^2, d^2 m_y))$. In our experiments, the feature dimension $d$ is typically much larger than the number of high-confidence samples $m_y$ per category in a batch, so the SVD complexity can be approximated as $O(dm_y^2)$; (3) Normalization and entropy computation only require $O(m_y)$, which is much lower than the SVD cost. Thus, the additional time complexity of DDDE per batch is dominated by $K$ SVD computations. The total additional overhead is approximately $O(d \sum_{y=1}^{K} m_y^2)$.

**LMC Time Complexity.** Let $C_f$ be the cost of a forward pass to compute Logits. The LMC process involves resampling the labeled set once on the estimated $\hat{P}_{Y_u}$ to obtain the proxy set $\mathcal{D}_v$, and then optimizing the scalar $\tau$ using cross-entropy loss: (1) Computing sampling probabilities for each sample and performing one Bernoulli sampling requires constant time, with time complexity $O(N)$; (2) Recomputing Logits for samples in $\mathcal{D}_v$ and adding the linear LA term $-\tau \cdot \hat{P}_{Y_u}$. This process includes, for each sample, one forward propagation ($O(C_f)$) and one $K$-dimensional $\mathrm{softmax}$ and cross-entropy computation ($O(K)$), totaling $O(V \cdot (C_f + K))$; (3) Gradient computation for the scalar and SGD update of $\tau$ both require constant time. Therefore, one LMC call only involves performing simple arithmetic operations on $K$-dimensional vectors (*i.e.*, the Logits) over $V$ samples to optimize the scalar, with an overall complexity of $O(N + V \cdot (C_f + K))$. This optimization is computationally trivial, consistent with the conclusion at the end of Appendix F.

**Actual Run-time.** We compare the training time of different methods using a single RTX 4090 GPU on the large-scale dataset SIN-127, including image resolutions of $32 \times 32$ and $64 \times 64$. The results are shown in Table 10, from which we can observe that the actual runtime of our method is not significantly different from the baseline method without DDDE and LMC (FixMatch+ACR).

Table 10: Actual run-time on SIN-127.

| Method | $32 \times 32$ | $64 \times 64$ |
|---|---|---|
| FixMatch | $31.12h$ | $84.05h$ |
| FixMatch+ACR | $38.27h$ | $96.62h$ |
| FixMatch+CoLA (Ours) | $39.53h$ | $97.63h$ |

## I  MORE COMPARISON RESULTS ON STL-10-LT

We evaluate the performance of CoLA versus ADELLO. The results are shown in Table 11, from which we can observe that: (1) ADELLO outperforms CoLA under the first, second, and fourth settings, which may be because the STL-10-LT dataset contains a large number of OOD samples in the unlabeled set, and the low-confidence distillation in ADELLO effectively acts as a threshold-based OOD filtering or regularization mechanism. It prevents the model from confidently and incorrectly assigning OOD samples to labeled categories or leveraging them for representation learning without hard label assignments, which is particularly advantageous on STL-10-LT. (2) When ADELLO does not employ distillation techniques (ADELLO*), focusing the comparison on handling imbalance, CoLA outperforms ADELLO*. This demonstrates that CoLA offers a superior solution for the core long-tail problem. The contributions of these two approaches are largely orthogonal.

Table 11: Comparison with ADELLO on STL-10-LT. $*$ represents w/o distillation.

| $(N_1, M)$ | $(150, \approx 100k)$ | | $(450, \approx 100k)$ | |
|---|---|---|---|---|
| $(\gamma_l, \gamma_u)$ | $(10, -)$ | $(20, -)$ | $(10, -)$ | $(20, -)$ |
| ADELLO* | $71.26 \pm 0.94$ | $67.36 \pm 1.37$ | $77.76 \pm 0.95$ | $75.17 \pm 1.06$ |
| ADELLO | $\mathbf{74.24 \pm 0.65}$ | $\mathbf{71.56 \pm 0.74}$ | $\underline{79.44 \pm 0.46}$ | $\mathbf{77.95 \pm 0.73}$ |
| CoLA (Ours) | $\underline{73.32 \pm 0.73}$ | $\underline{68.96 \pm 2.55}$ | $\mathbf{79.70 \pm 0.48}$ | $\underline{77.53 \pm 0.80}$ |

## J BREAKDOWN OF RESULTS ON CIFAR-10/100-LT

The comprehensive breakdown of comparison results on CIFAR-10/100-LT under consistent, uniform, reversed, middle, and head-tail distributions is shown in Table 12, 13, 14, 15, and 16, respectively. Our method achieves SOTA performance across almost all data distributions.

Table 12: Comparison with other LTSSL methods under the consistent distribution. Top and second-best performances are bolded and underlined, respectively. Metrics represent mean accuracy and standard deviation over five independent runs.

| Dataset | CIFAR-10-LT | | | | CIFAR-100-LT | |
|---|---|---|---|---|---|---|
| $(N_1, M_1)$ | (1500, 3000) | | (500, 4000) | | (150, 300) | |
| $\gamma_l = \gamma_u$ | 100 | 150 | 100 | 150 | 10 | 15 |
| Supervised | 63.72±1.51 | 59.61±1.58 | 46.84±1.23 | 42.92±1.14 | 48.59±0.64 | 45.66±0.57 |
| FixMatch | 76.71±1.00 | 72.63±0.80 | 72.13±2.12 | 66.25±1.53 | 57.62±0.73 | 54.16±0.65 |
| *Logit Adjustment-Based* | | | | | | |
| ACR | 84.17±0.50 | 81.20±0.77 | 81.31±0.93 | 76.86±2.15 | 59.85±0.66 | 56.76±0.81 |
| CPE | 84.16±0.41 | 81.28±0.56 | 79.91±1.05 | 75.76±0.91 | 55.14±0.43 | 51.09±0.59 |
| Meta-Expert | 83.92±0.46 | 82.07±0.34 | 81.77±1.21 | 77.54±1.16 | 56.49±1.01 | 52.71±1.00 |
| Sim-Pro | 84.04±0.45 | 82.21±0.15 | **81.91±0.70** | 78.11±0.95 | 59.69±0.51 | 56.90±0.81 |
| TRAS | 66.40±0.98 | 62.23±1.06 | 56.73±1.09 | 51.46±1.46 | 51.48±0.66 | 47.55±0.51 |
| *Loss Reweighting-Based* | | | | | | |
| DeBiasPL | 77.78±1.05 | 73.37±0.73 | 72.71±1.28 | 66.17±1.70 | 57.90±0.68 | 54.84±0.73 |
| SAW | 80.75±0.69 | 77.33±0.64 | 77.13±1.31 | 71.09±1.68 | 58.59±0.56 | 55.59±0.82 |
| *Resampling-Based* | | | | | | |
| ABC | 83.62±0.80 | 80.62±0.57 | 79.58±0.63 | 76.78±0.32 | 58.50±0.40 | 55.63±0.42 |
| ADSH | 77.92±0.82 | 73.20±0.47 | 75.22±0.83 | 68.04±1.93 | 58.12±0.44 | 54.85±0.81 |
| CReST | 79.04±0.49 | 74.78±0.61 | 77.14±1.88 | 69.68±2.12 | 57.76±0.32 | 54.60±0.47 |
| CReST+ | 79.19±0.44 | 74.61±0.44 | 78.66±0.44 | 71.96±1.60 | 57.51±0.31 | 54.56±0.40 |
| *Data Mixing-Based* | | | | | | |
| BEM | 80.12±0.78 | 75.71±0.60 | 75.32±0.74 | 69.92±0.52 | 59.03±0.95 | 56.13±0.36 |
| CoSSL | 83.41±1.18 | 81.63±0.26 | 80.29±0.67 | 76.74±0.32 | 58.06±0.65 | 55.74±0.45 |
| *Distribution Alignment-Based* | | | | | | |
| DARP | 78.25±0.54 | 74.43±0.44 | 74.75±1.23 | 69.00±0.76 | 58.21±0.57 | 55.24±0.61 |
| RDA | 77.42±1.32 | 73.21±1.25 | 73.48±1.09 | 68.45±0.65 | 58.13±0.52 | 54.51±0.67 |
| *Others* | | | | | | |
| DASO | 75.84±0.72 | 72.21±0.73 | 69.15±1.10 | 63.46±1.24 | 58.95±0.51 | 55.69±0.74 |
| CoLA (Ours) | **85.04±0.32** | **82.42±0.54** | 81.78±0.55 | **78.24±2.09** | **60.42±0.60** | **57.67±0.80** |

Table 13: Comparison with other LTSSL methods under the uniform distribution. Top and second-best performances are bolded and underlined, respectively. Metrics represent mean accuracy and standard deviation over five independent runs.

| Dataset | CIFAR-10-LT | | CIFAR-100-LT | |
|---|---|---|---|---|
| $(N_1, M)$ | (1500, 3000) | | (150, 300) | |
| $\gamma_l$ | 100 | 150 | 10 | 15 |
| Supervised | 63.46±1.07 | 58.90±1.63 | 48.61±0.55 | 45.57±0.36 |
| FixMatch | 73.11±1.25 | 69.36±2.14 | 48.93±0.51 | 45.88±0.30 |
| *Logit Adjustment-Based* | | | | |
| ACR | 83.04±0.32 | 81.28±0.54 | 50.24±0.69 | 47.72±0.50 |
| CPE | 83.45±0.58 | 81.73±0.82 | 48.13±0.60 | 45.64±0.78 |
| Meta-Expert | 84.04±0.48 | 82.21±0.60 | 49.08±0.88 | 46.61±0.93 |
| Sim-Pro | 81.40±0.72 | 80.32±0.54 | 49.62±0.94 | 46.51±0.48 |
| TRAS | 60.28±0.73 | 56.22±1.04 | 46.90±0.54 | 43.24±0.56 |
| *Loss Reweighting-Based* | | | | |
| DeBiasPL | 74.00±1.39 | 69.29±2.51 | 49.43±0.55 | 46.42±0.56 |
| SAW | 81.84±0.20 | 80.85±0.65 | 49.63±0.66 | 46.76±0.59 |
| *Resampling-Based* | | | | |
| ABC | 81.70±0.64 | 80.47±0.61 | 50.46±0.37 | 47.68±0.46 |
| ADSH | 76.35±0.83 | 73.97±0.80 | 49.74±0.86 | 46.62±0.34 |
| CReST | 83.26±0.83 | 80.73±0.87 | 49.03±0.27 | 45.96±0.23 |
| CReST+ | 83.02±0.73 | 79.99±0.34 | 49.23±0.34 | 46.02±0.16 |
| *Data Mixing-Based* | | | | |
| BEM | 81.35±0.82 | 78.79±0.72 | 50.27±0.73 | 48.16±0.58 |
| CoSSL | 80.51±1.07 | 78.33±1.40 | 50.79±0.50 | 48.34±0.44 |
| *Distribution Alignment-Based* | | | | |
| DARP | 73.49±1.39 | 69.69±2.39 | 49.29±0.50 | 46.35±0.30 |
| RDA | 78.08±0.60 | 76.95±1.12 | 49.30±0.64 | 45.98±0.30 |
| *Others* | | | | |
| DASO | 76.22±1.23 | 73.27±1.22 | 49.33±0.55 | 46.44±0.43 |
| CoLA (Ours) | **84.83±0.34** | **82.48±0.39** | **51.28±0.45** | **49.23±0.77** |

Table 14: Comparison with other LTSSL methods under the reversed distribution. Top and second-best performances are bolded and underlined, respectively. Metrics represent mean accuracy and standard deviation over five independent runs.

| Dataset | CIFAR-10-LT | | CIFAR-100-LT | |
|---|---|---|---|---|
| $(N_1, M_K)$ | (1500, 3000) | | (150, 300) | |
| $\gamma_l = \gamma_u$ | 100 | 150 | 10 | 15 |
| Supervised | 63.76±1.40 | 59.71±1.56 | 48.48±0.33 | 45.42±0.24 |
| FixMatch | 70.77±2.19 | 65.82±1.46 | 57.70±0.52 | 53.97±0.59 |
| *Logit Adjustment-Based* | | | | |
| ACR | 84.35±1.01 | 82.62±0.27 | 60.27±0.46 | 58.16±0.33 |
| CPE | 84.46±1.09 | 83.56±0.45 | 59.64±0.28 | 57.58±0.31 |
| Meta-Expert | 85.36±0.24 | **84.69±0.56** | 60.14±0.64 | 57.04±0.63 |
| Sim-Pro | 75.83±0.82 | 73.60±1.33 | 59.07±0.52 | 55.66±0.71 |
| TRAS | 57.93±0.68 | 54.39±0.98 | 48.37±0.47 | 43.78±0.50 |
| *Loss Reweighting-Based* | | | | |
| DeBiasPL | 71.44±2.06 | 65.92±2.05 | 58.47±0.61 | 54.73±0.82 |
| SAW | 76.09±0.84 | 73.56±0.89 | 59.03±0.63 | 55.98±0.70 |
| *Resampling-Based* | | | | |
| ABC | 82.56±0.92 | 81.10±0.30 | 59.39±0.63 | 57.57±0.67 |
| ADSH | 69.95±0.86 | 67.43±1.45 | 56.07±0.65 | 52.64±0.36 |
| CReST | 84.26±0.70 | 82.95±3.83 | 59.31±0.58 | 56.59±0.47 |
| CReST+ | 72.17±1.25 | 67.08±0.63 | 59.03±0.29 | 55.70±0.52 |
| *Data Mixing-Based* | | | | |
| BEM | 79.27±1.20 | 76.27±1.26 | 59.54±0.79 | 57.06±1.35 |
| CoSSL | 76.20±1.88 | 72.56±0.69 | 58.83±0.51 | 55.63±0.80 |
| *Distribution Alignment-Based* | | | | |
| DARP | 70.96±2.47 | 66.73±1.94 | 58.04±0.59 | 54.75±1.04 |
| RDA | 73.05±0.51 | 70.56±0.42 | 57.82±0.57 | 54.50±0.75 |
| *Others* | | | | |
| DASO | 77.61±1.05 | 75.33±1.51 | 59.64±0.71 | 56.22±0.95 |
| CoLA (Ours) | **86.84±0.73** | 84.38±1.08 | **61.40±0.65** | **59.38±0.59** |

Table 15: Comparison with other LTSSL methods under the middle distribution. Top and second-best performances are bolded and underlined, respectively. Metrics represent mean accuracy and standard deviation over five independent runs.

| Dataset | CIFAR-10-LT | | | | CIFAR-100-LT | |
|---|---|---|---|---|---|---|
| $(N_1, M_{\lceil K/2 \rceil})$ | (1500, 3000) | | (500, 4000) | | (150, 300) | |
| $\gamma_l = \gamma_u$ | 100 | 150 | 100 | 150 | 10 | 15 |
| Supervised | 63.76±1.38 | 59.38±1.54 | 46.82±2.20 | 43.94±1.52 | 48.24±0.68 | 45.41±0.37 |
| FixMatch | 75.65±0.71 | 71.63±0.78 | 68.76±1.43 | 64.36±1.36 | 57.49±0.35 | 53.81±0.70 |
| *Logit Adjustment-Based* | | | | | | |
| ACR | 84.06±0.81 | 81.86±1.06 | 80.73±0.68 | 72.32±2.48 | 59.13±0.32 | 56.18±0.70 |
| CPE | 84.42±0.61 | 81.98±0.35 | 80.17±1.34 | 76.65±1.35 | 56.53±0.56 | 52.31±0.98 |
| Meta-Expert | 84.36±0.21 | 82.26±0.37 | 80.71±0.70 | 76.07±1.26 | 57.00±0.51 | 53.18±0.62 |
| Sim-Pro | 82.54±0.41 | 79.75±0.87 | 76.75±1.12 | 71.72±1.38 | 59.05±0.65 | 56.05±0.60 |
| TRAS | 62.74±0.64 | 57.71±0.97 | 44.75±1.32 | 40.15±2.12 | 50.04±0.71 | 45.24±0.23 |
| *Loss Reweighting-Based* | | | | | | |
| DeBiasPL | 76.83±0.46 | 72.27±0.33 | 68.92±0.82 | 64.68±1.30 | 57.69±0.83 | 54.62±0.40 |
| SAW | 80.09±0.56 | 76.34±0.92 | 73.54±2.00 | 67.58±2.80 | 58.66±0.66 | 55.19±0.65 |
| *Resampling-Based* | | | | | | |
| ABC | 83.23±0.54 | 80.56±0.80 | 78.61±0.61 | 72.40±2.07 | 58.81±0.62 | 55.80±0.68 |
| ADSH | 72.84±0.87 | 68.04±1.02 | 62.51±0.83 | 58.41±1.55 | 56.56±0.48 | 52.89±0.67 |
| CReST | 83.73±0.25 | 79.86±0.56 | 81.66±0.37 | 73.86±2.92 | 58.71±0.23 | 56.22±0.53 |
| CReST+ | 81.82±0.95 | 76.76±0.76 | 81.81±0.80 | 72.98±2.30 | 58.48±0.39 | 55.14±0.22 |
| *Data Mixing-Based* | | | | | | |
| BEM | 84.35±0.37 | 82.07±0.42 | 79.12±1.02 | 75.62±1.20 | 59.11±0.37 | 55.80±0.52 |
| CoSSL | 83.39±0.37 | 81.04±0.44 | 78.17±1.03 | 74.67±1.29 | 58.65±0.50 | 55.95±0.77 |
| *Distribution Alignment-Based* | | | | | | |
| DARP | 76.65±0.60 | 72.34±0.58 | 68.90±0.77 | 65.32±1.35 | 57.63±0.49 | 54.02±0.71 |
| RDA | 70.93±0.55 | 66.73±0.68 | 62.66±1.59 | 57.98±1.52 | 57.70±0.33 | 53.37±0.42 |
| *Others* | | | | | | |
| DASO | 75.20±1.05 | 70.96±0.92 | 65.93±1.94 | 61.18±1.89 | 58.91±0.74 | 55.64±0.61 |
| CoLA (Ours) | **85.16±0.73** | **83.27±0.37** | **82.07±0.45** | **76.94±2.81** | **60.10±0.41** | **57.32±0.76** |

Table 16: Comparison with other LTSSL methods under the head-tail distribution. Top and second-best performances are bolded and underlined, respectively. Metrics represent mean accuracy and standard deviation over five independent runs.

| Dataset | CIFAR-10-LT | | | | CIFAR-100-LT | |
|---|---|---|---|---|---|---|
| $(N_1, M_{K^{(K+1) \bmod 2}})$ | (1500, 3000) | | (500, 4000) | | (150, 300) | |
| $\gamma_l = \gamma_u$ | 100 | 150 | 100 | 150 | 10 | 15 |
| Supervised | 64.03±1.48 | 59.73±1.78 | 47.17±2.04 | 43.61±1.68 | 48.49±0.34 | 45.52±0.20 |
| FixMatch | 75.13±1.74 | 68.99±2.07 | 63.87±3.09 | 58.00±2.37 | 58.25±0.72 | 54.93±1.01 |
| *Logit Adjustment-Based* | | | | | | |
| ACR | 83.48±0.47 | 81.42±0.57 | 76.13±0.98 | 72.64±0.90 | 59.93±0.53 | 57.58±0.38 |
| CPE | 83.36±0.24 | 81.56±0.09 | 78.53±0.42 | 75.06±4.58 | 58.57±0.68 | 55.74±0.17 |
| Meta-Expert | 83.70±0.92 | 81.79±0.98 | 78.89±0.53 | 75.08±1.13 | 59.22±0.52 | 56.06±0.63 |
| Sim-Pro | 79.72±0.73 | 77.50±1.20 | 76.38±2.51 | 73.54±2.86 | 59.72±0.74 | 57.00±0.63 |
| TRAS | 59.09±0.59 | 55.81±1.17 | 49.08±0.58 | 45.47±1.00 | 50.48±1.01 | 45.79±0.82 |
| *Loss Reweighting-Based* | | | | | | |
| DeBiasPL | 75.98±1.67 | 69.58±4.04 | 64.69±3.56 | 58.97±2.78 | 58.44±0.81 | 55.52±0.69 |
| SAW | 79.06±0.90 | 77.93±1.23 | 76.54±0.57 | 73.28±1.13 | 59.43±0.81 | 56.32±0.92 |
| *Resampling-Based* | | | | | | |
| ABC | 82.98±0.33 | 81.04±0.76 | 78.87±0.34 | 74.71±0.95 | 60.17±0.47 | 57.32±0.48 |
| ADSH | 75.51±1.81 | 73.92±1.07 | 74.55±2.10 | 70.68±1.12 | 58.15±0.39 | 54.38±0.81 |
| CReST | 83.29±0.28 | 82.00±0.49 | 74.64±4.67 | 70.89±3.64 | 57.83±0.23 | 55.04±0.32 |
| CReST+ | 80.39±0.87 | 75.25±2.97 | 72.44±2.73 | 64.12±0.89 | 57.90±0.20 | 55.25±0.29 |
| *Data Mixing-Based* | | | | | | |
| BEM | 78.45±1.32 | 75.01±2.77 | 72.08±2.21 | 67.93±1.21 | 59.28±0.51 | 56.83±1.07 |
| CoSSL | 77.59±1.36 | 74.17±2.47 | 71.43±2.08 | 67.00±1.41 | 59.48±0.59 | 56.63±0.96 |
| *Distribution Alignment-Based* | | | | | | |
| DARP | 75.87±2.02 | 69.50±2.23 | 63.66±2.25 | 59.29±3.36 | 58.50±0.60 | 55.35±1.02 |
| RDA | 75.42±0.85 | 73.32±1.05 | 69.00±1.52 | 65.60±0.76 | 58.80±0.75 | 55.31±0.74 |
| *Others* | | | | | | |
| DASO | 75.11±1.26 | 71.41±1.69 | 66.21±1.11 | 61.70±1.77 | 59.86±0.73 | 56.45±0.70 |
| CoLA (Ours) | **84.42±0.45** | **82.51±0.28** | **79.80±0.34** | **75.88±0.59** | **61.17±0.56** | **58.61±0.54** |

