# OpenReview forum: "CoLA: Co-Calibrated Logit Adjustment for Long-Tailed Semi-Supervised Learning"
_ICLR.cc/2026/Conference — ICLR 2026 Poster_

### Official Review · Reviewer_DkFy · 2025-10-28

**Soundness:** 2
**Presentation:** 2
**Contribution:** 3
**Rating:** 6
**Confidence:** 2

**Summary:**

This paper addresses the issue of pseudo‑label bias in long‑tailed semi‑supervised learning, and identifies two core limitations in existing Logit Adjustment (LA)–based approaches: distribution estimation distortion caused by sample redundancy, and the use of a fixed overall adjustment strength. To overcome these limitations, the authors propose the CoLA framework, whose main components include: estimating the effective number of samples via the effective rank of the representation matrix to obtain a de‑duplicated class distribution; and, based on the estimated distribution, constructing a proxy validation set and optimizing the overall adjustment strength through meta‑learning so that it adapts to the characteristics of the current distribution. Furthermore, the paper provides a theoretical generalization error bound for the proposed LMC and validates the method through experiments on benchmark datasets.

**Strengths:**

1. The problems identified in the paper are reasonable and important, and the authors provide a detailed analysis supported by both theoretical justification and experimental evidence.
2. The paper presents a generalization bound and a convexity analysis for the meta‑learning process, which enhances the theoretical soundness of the proposed method.
3. The visualization in the ablation study is presented in a clear and comprehensive manner.

**Weaknesses:**

1. The computation of effective rank and the meta‑learning procedure may introduce substantial computational overhead. It would be beneficial to include an analysis of the time complexity and computational complexity, particularly with respect to runtime performance on large‑scale datasets.
2. In the downstream experimental evaluation, I notice that the SIN‑127 dataset is a down‑sampled version of ImageNet‑127. Why not perform testing directly on ImageNet‑127? I am curious about the potential results on the full ImageNet‑127 dataset.
3. The paper lacks a primary diagram illustrating the proposed method. Introducing a main figure would make the methodology clearer and facilitate reader understanding.

**Questions:**

See the weaknesses.

---

> ### Author Response · Authors · 2025-11-20
> **Responses to W1-W2**
>
> Thank you for your insightful feedback and questions. We appreciate the opportunity to clarify these important points about our work.
>
> **W1: The computation of effective rank and the meta‑learning procedure may introduce substantial computational overhead. It would be beneficial to include an analysis of the time complexity and computational complexity, particularly with respect to runtime performance on large‑scale datasets.**
>
> **Response:** This is a crucial point. We have included a detailed complexity analysis and a comparison of actual runtimes in Appendix H of the revised manuscript.
>
> **DDDE Time Complexity**
> Let $K$ be the number of categories, $d$ the feature dimension, and $m _ y$ the number of high-confidence samples for category $y \in \mathcal{Y}$ in the current iteration, with $\mathcal{M} = \sum _ y m _ y$ (the total number of high-confidence samples in each batch, not exceeding the unlabeled batch size $|\mathcal{B} _ u|$).
>
> In each batch, DDDE first stacks high-confidence unlabeled features for each category $y$ into a matrix $\pmb{Z} _ y \in \mathbb{R}^{d \times m _ y}$, then performs singular value decomposition (SVD) and computes entropy and $\mathrm{erank}$:
> 1. Feature collection and matrix formation: Each sample is copied once, resulting in a cost of $O(d\mathcal{M})$;
> 2. SVD: Performing full SVD on $\pmb{Z} _ y$ costs $O(\min(dm _ y^2, d^2m _ y))$. In our experiments, the feature dimension $d$ is typically much larger than the number of high-confidence samples $m _ y$ per category in a batch, so the SVD complexity can be approximated as $O(dm _ y^2)$.
> 3. Singular value normalization: Normalization and entropy calculation require only $O(m _ y)$, which is much lower than the SVD computation cost.
>
> Thus, the additional time complexity of DDDE per batch is dominated by $K$ SVD computations. The total additional overhead is approximately $O(d \sum _ {y=1}^{K} m _ y^2)$.
>
> **LMC Time Complexity**
> Let $N$ be the total number of labeled samples, $V$ the number of samples expected to be sampled into $\mathcal{D} _ v$, and $C _ f$ the cost of a forward pass to compute logits.
>
> The LMC process involves resampling the labeled set once on the estimated $\hat{P} _ {Y _ u}$ to obtain the proxy set $\mathcal{D} _ v$, then optimizing the scalar $\tau$ using cross-entropy:
> 1. Calculating sampling probabilities based on class ratios and performing Bernoulli sampling: Each sample requires constant time for probability computation and one random trial, resulting in $O(N)$.
> 2. Forward pass and cross-entropy summation: Recomputing logits for samples in $\mathcal{D} _ v$ and adding the linear LA term $-\tau\cdot\hat{P} _ {Y _ u}$: Each sample requires one forward pass ($O(C _ f)$) and one $K$-dimensional softmax/cross-entropy computation ($O(K)$), totaling $O(V \cdot (C _ f + K))$.
> 3. Updating $\tau$: Gradient computation with respect to the scalar and SGD update are both constant time.
>
> Thus, the overall complexity of an LMC is $O\left(N + V \cdot (C _ f + K)\right)$.
> This overhead is minimal, as it primarily involves basic arithmetic operations on a $K$-dimensional vector (logits) across $V$ samples to optimize a scalar. This aligns with the conclusion at the end of Appendix F, stating that this optimization is computationally trivial.
>
> **Actual Runtime**
> We compare the training time of different methods using a single RTX 4090 GPU on the large-scale dataset SIN-127 (including image resolutions of $32 \times 32$ and $64 \times 64$). As shown below, the runtime of our method is comparable to the baseline (FixMatch+ACR), with only a marginal increase due to DDDE and LMC.
>
> | Methods                | $32\times32$ | $64\times64$ |
> | --------------         | ------------ | ------------ |
> | FixMatch               | $31.12h$     | $84.05h$     |
> | FixMatch+ACR           | $38.27h$     | $96.62h$     |
> | FixMatch+CoLA (Ours)   | $39.53h$     | $97.63h$     |
> |  |  |  |
>
> **W2: In the downstream experimental evaluation, I notice that the SIN‑127 dataset is a down‑sampled version of ImageNet‑127. Why not perform testing directly on ImageNet‑127? I am curious about the potential results on the full ImageNet‑127 dataset.**
>
> **Response:** Thank you for this suggestion. We select SIN-127 primarily to maintain consistency with established benchmarks in the field. Most recent LTSSL studies [1-6] evaluate performance on the down-sampled version of ImageNet-127 to ensure fair and direct comparisons. Due to computational constraints within the rebuttal timeframe, we are unable to complete experiments on the full-resolution ImageNet-127. However, we plan to include these results in the final version.

---

> > ### Author Response · Authors · 2025-11-20
> > **Response to W3**
> >
> > **W3: The paper lacks a primary diagram illustrating the proposed method. Introducing a main figure would make the methodology clearer and facilitate reader understanding.**
> >
> > **Response:** We appreciate this suggestion, as it has significantly helped us improve the clarity of our presentation.
> > We have provided the pseudocode in the revised manuscript (Page 21, Algorithm 1) to illustrate the overall workflow of our method and will add a main figure in the final version.
> >
> > References:
> > [1] Fan et al., CoSSL: Co-Learning of Representation and Classifier for Imbalanced Semi-Supervised Learning, CVPR 2022.
> > [2] Wei et al., Towards Realistic Long-Tailed Semi-Supervised Learning: Consistency Is All You Need, CVPR 2023.
> > [3] Du et al., SimPro: A Simple Probabilistic Framework Towards Realistic Long-Tailed Semi-Supervised Learning, ICML 2024.
> > [4] Ma et al., Three Heads Are Better than One: Complementary Experts for Long-Tailed Semi-supervised Learning, AAAI 2024.
> > [5] Zheng et al., BEM: Balanced and Entropy-based Mix for Long-Tailed Semi-Supervised Learning, CVPR 2024.
> > [6] Hou et al., A Square Peg in a Square Hole: Meta-Expert for Long-Tailed Semi-Supervised Learning, ICML 2025.

---

### Official Review · Reviewer_peuh · 2025-10-29

**Soundness:** 2
**Presentation:** 3
**Contribution:** 2
**Rating:** 4
**Confidence:** 4

**Summary:**

This paper attempts to tackle an problem in Long-Tailed Semi-Supervised Learning (LTSSL): the confirmation bias driven by biased pseudo-labels. The authors claim that existing methods based on LA suffer from two critical limitations: 1) they rely on naive frequency counting to estimate the unlabeled data distribution, and 2) they treat the overall adjustment strength as a fixed hyperparameter. The authors propose CoLA, which consists of two main components: first, a De-Duplicated Distribution Estimation (DDDE) module that attempts to estimate a more accurate class distribution by calculating the effective rank of class representations to account for sample redundancy. Second, a Logit Meta-Calibration (LMC) procedure that constructs a proxy validation set and uses meta-learning to automatically optimize the overall adjustment strength. The experiment results show that their method establishes a new state-of-the-art on four public benchmarks.

**Strengths:**

1. The authors do identify a potentially interesting aspect of the LA mechanism, namely the interplay between the class-wise adjustment and the overall adjustment strength. This is a reasonable observation.
2. The authors evaluate their method across multiple datasets and distribution mismatch scenarios.

**Weaknesses:**

1. The first of my concern is novelty. The core idea of this paper is little more than a combination of existing techniques, such as effective number, dual-branch, and meta-learning for hyperparams.
2. The description of the DDDE module is overly simplistic. The authors propose computing the effective rank for the representation matrix Zy of each class y. They fail to discuss the computational cost of this procedure.
3. In Figure2 (b,d,e), after applying LMC, the slope of the pseudo-label accuracy improvement barely changes. So I think its contribution is questionable.
4. The proxy validation set Dv is resampled from the small and imbalanced labeled set Dl. For tail classes, Dl may contain only a few samples. How do you justify that such a severely constrained proxy set can effectively guide the learning of $\tau$ for a massive and differently distributed unlabeled set Du?

**Questions:**

See in Weaknesses.

---

> ### Author Response · Authors · 2025-11-20
> **Responses**
>
> Thank you very much for your questions. We hope our responses can address all your concerns.
>
> **W1: The first of my concern is novelty. The core idea of this paper is little more than a combination of existing techniques, such as effective number, dual-branch, and meta-learning for hyperparams.**
>
> **Response:** We respectfully disagree that our approach is a simple, heuristic combination of existing techniques. Every design aspect is theoretically grounded.
>
> First, Proposition 1 (Section 5) provides a generalization upper bound for LMC, proving that minimizing the empirical proxy risk $\hat{R} _ {\mathcal{D} _ v}(h _ \tau)$ bounds the true target risk $R _ {P _ u}(h _ \tau)$.
> Second, we adopt the linear LA term $-\tau\cdot\pmb{p}$ to ensure convexity of the LMC's optimization objective (Appendix F), enabling standard optimization methods (like gradient descent) to efficiently and reliably converge to the unique global minimum $\tau^*$.
> Third, accurate estimation of $\hat{P} _ {Y _ u}$ by DDDE leads to tighter generalization bounds (see our responses to Weakness 2, Reviewer nQAr).
>
> In summary, our core contribution lies in identifying and solving a critical, unaddressed coupling problem within the LA mechanism itself.
>
> **W2: The description of the DDDE module is overly simplistic. The authors propose computing the effective rank for the representation matrix Zy of each class y. They fail to discuss the computational cost of this procedure.**
>
> **Response:** This is a crucial point, and we apologize for not making the cost analysis in the main text. We provide a detailed analysis in our response to **Reviewer DkFy Weakness 1** and have included it in the Appendix H in the revised version.
>
> **W3: In Figure2 (b,d,e), after applying LMC, the slope of the pseudo-label accuracy improvement barely changes. So I think its contribution is questionable.**
>
> **Response:** Thank you for this sharp observation. Indeed, while the experiments on CIFAR-10-LT in Figure 2(b,d,e) show a significant improvement in pseudo-label accuracy after adopting LMC (after the epoch indicated by the gray dashed line), on CIFAR-100-LT, it merely maintains the pre-adoption upward trend of pseudo-label accuracy.
> This may be because pseudo-label accuracy is calculated based on unlabeled **training set**, while the ultimate goal is generalization.
> As shown in the ablation study (Table 4), the final **test set accuracy** using LMC (w/o D-L) is significantly higher than using any fixed $\tau$ (w/o D-$1,2,4$). This confirms that while the training dynamics might look similar visually, LMC learns a $\tau^*$ that yields a decision boundary with superior generalization capability.
>
> **W4: The proxy validation set Dv is resampled from the small and imbalanced labeled set Dl. For tail classes, Dl may contain only a few samples. How do you justify that such a severely constrained proxy set can effectively guide the learning of $\tau$ for a massive and differently distributed unlabeled set Du?**
>
> **Response:** This is a fundamental and excellent question. The justification for using a small, resampled proxy set $\mathcal{D} _ v$ is four-fold:
> 1. A Highly Constrained Task: The proxy set $\mathcal{D} _ v$ is not used to train the deep backbone (which uses all of $\mathcal{D} _ l$ and $\mathcal{D} _ u$). Its sole purpose is to optimize one single scalar parameter $\tau$.
> 2. Convex Optimization: As we formally state and analyze in Appendix F, the optimization objective $\mathcal{L}(\tau)$ is convex with respect to $\tau$. Because this is a simple, $1$-dimensional convex problem, it does not suffer from spurious local minima and does not require a massive dataset to robustly find the global optimum $\tau^*$.
> 3. Distribution Matching is Key: The critical property of $\mathcal{D} _ v$ is not its size, but that its class distribution is constructed to match our best estimate of the unlabeled distribution $\hat{P} _ {Y _ u}(y)$.
> We achieve this by resampling $\mathcal{D} _ l$ using the DDDE-estimated probabilities. This ensures that $\mathcal{D} _ v$ is a faithful proxy of the target distribution's class proportions.
> 4. Theoretical Justification: Our generalization analysis (Section 5) provides the formal guarantee. Proposition 1 bounds the true target risk $R _ {P _ u}(h _ \tau)$ by the empirical proxy risk $\hat{R} _ {\mathcal{D} _ v}(h _ \tau)$. The tightness of this bound is governed by the discrepancy term $|\hat{R} _ {\mathcal{D} _ v,w}-\hat{R} _ {\mathcal{D} _ v}|$, which, as we explained to Reviewer nQAr (Weakness 1-2), shrinks as our distribution estimate $\hat{P} _ {Y _ u}$ becomes more accurate.
>
> Therefore, by first using DDDE to get an accurate estimate and then constructing $\mathcal{D} _ v$ to match it, we ensure $\mathcal{D} _ v$ is a small but sufficient and theoretically-grounded set for its specific, convex task.

---

> > ### Comment · Reviewer_peuh · 2025-11-25
> >
> > I have read the rebuttal and raise my score accordingly.

---

### Official Review · Reviewer_bnRy · 2025-10-30

**Soundness:** 2
**Presentation:** 3
**Contribution:** 2
**Rating:** 4
**Confidence:** 4

**Summary:**

The authors address in their manuscript the problem of long-tailed semi-supervised learning. They analyze the weaknesses of several methods based on logit adjustment and propose an approach CoLA that is claimed to suffer less from over-suppression. They further propose DDDE to estimate the unlabeled distribution and LMC as a meta-learning strategy. They validate their method on 4 different datasets (CIFAR-10/100-LT, STL-10-LT, and SIN-127).

**Strengths:**

1. The paper is well-written (with exception line 090 "each class's representations") and structured (DDDE=4.1, LMC=4.2, CoLA=4.3).
2. The main hypothesis is clearly stated and the approach is presented in a plausible way.
3. Technical parts are accurately and detailed described.
4. A good set of datasets has been selected for the experiments.
5. The results are good (even if not always beating the state of the art)

**Weaknesses:**

1. The related work is covering a subset of the field, important references (such as [1]) are only used in the appendix, other are completely missing e.g. [2-5].
2. The premise for the main hypothesis, i.e., the negative effect of overlooking the interplay between the two types of adjustment, has not been clearly supported by experimental results. If the premise is not properly established, all subsequent claims are affected. Figure 1b) only shows that monotonicity is not fulfilled.
3. Some statements lack evidence or reference, e.g. line 085. "current work" embraces all works, see also 1. (this is not saying that any of [2-5] does).
4. The protocol (for CIFAR) chosen according to Du et al. ICML 2024 (Simpro) deviates from other, previously published protocols, e.g. from the cited paper [6], and makes comparisons difficult, in particular to state-of-the-art methods that remained un-cited and that use those previous protocols.
5. For experiments that use compatible protocols, e.g. STL-10-LT, the proposed method is inferior to e.g. [4} and rather en par with [3].
6. The description of erank is not sufficiently self-contained. In particular, it is not clear why the EN is estimated by erank. Line 197 just says "we quantify EN ... using ... erank" and references point to EN and erank, but not why the quantification is possible.  Also, it is not obvious why line 205 $p(i)$ is a probability and not just a point in the $m_y$-simplex.
7. 4.2 is partly written in a procedural way and the overall approach that leads to the algorithm needs to be stated more clearly.

[1] Zhang et al. Mixup: Beyond empirical risk minimization. ICLR 2018.

[2] Lazarow et al. Unifying distribution alignment as a loss for imbalanced semi-supervised learning. CVPR 2023.

[3] Chen et al. Softmatch: Addressing the quantity-quality tradeoff in semi-supervised learning. ICLR 2023.

[4] Aimar et al. Flexible distribution alignment: Towards long-tailed semi-supervised learning with proper calibration. ECCV 2024.

[5] Kim et al. Separated and Independent Contrastive Semi-Supervised Learning for Imbalanced Datasets. IEEE Access 2025.

[6] Kim et al. Distribution aligning refinery of pseudo-label for imbalanced semi-supervised learning. NeurIPS 2020.

**Questions:**

1. (related to weakness 2.): which are the experimental results that clearly show that the _interplay_ of the two types of LA cause the issue?
2. (related to weakness 4.): what results are obtained with the protocol from Kim et al. 2020 or is there some other way to make the results comparable?
3. (related to weakness 5./1.): as the statement about state-of-the-art results need to be revised: in which situation does the proposed method shows its main strengths and weaknesses?
4. (related to weakness 6.): why is the quantification possible and why is $p(i)$ a proper probability?
5. (related to weakness 7.): what is the overall approach in 4.2?

---

> ### Author Response · Authors · 2025-11-20
> **Responses to W1-W3**
>
> Thank you very much for your careful comments. Several weaknesses and questions point to the same underlying issue. To avoid redundancy, we group these related comments and address them together in the following responses.
>
> **W1: The related work is covering a subset of the field, important references (such as [1]) are only used in the appendix, other are completely missing e.g. [2-5].**
>
> **Response:** Thank you for pointing out the insufficient coverage of related work. We have added discussions on works [2-5] in the related work sections (Section 2, Appendix B):
> UDAL [2] unifies Distribution Alignment (DA) and Logit Adjustment (LA), offering a loss-level perspective.
> SoftMatch [3], while designed for standard SSL, addresses the quantity-quality trade-off relevant to pseudo-labeling in LTSSL.
> ADELLO [4] and SICSSL [5] introduce dynamic alignment and decoupled pre-training, respectively.
>
> Since our work primarily focuses on improvements to LA techniques and due to page limitations, the main text's related work section only discusses most LA-based LTSSL works, while other types of work, such as Mixup [1] based on data mixing, are discussed in the appendix.
>
> **W2 (Q1): The premise for the main hypothesis, i.e., the negative effect of overlooking the interplay between the two types of adjustment, has not been clearly supported by experimental results. If the premise is not properly established, all subsequent claims are affected. Figure 1b only shows that monotonicity is not fulfilled. Which are the experimental results that clearly show that the interplay of the two types of LA cause the issue?**
>
> **Response:** Thanks for your question. The premise of our interplay hypothesis is that class-wise adjustment and the global scaling $\tau$ must be jointly adapted. Otherwise, pseudo-labels become systematically biased. This premise is supported not only by the intuition in Figure 1(b) but also by the ablation results in Section 6.3.
>
> **Fixed overall adjustment and ignoring interactions lead to significant performance loss.**
> In Table 4, under $10$ settings on CIFAR-10-LT and CIFAR-100-LT, the best performance among the three methods "w/o D-$\tau,\tau\in{1,2,4}$" (naive frequency estimation for class-wise adjustment + fixed overall adjustment) is still lower than "w/o D-L" (naive frequency estimation for class-wise adjustment + LMC-learned $\tau$). This difference is more pronounced on the CIFAR-100-LT dataset. For example, under the consistent long-tailed setting $(k _ {\max},k _ {\min})=(1,100)$, "w/o D-L" achieves $3.93\\%$ higher accuracy than the best fixed adjustment "w/o D-1", indicating that when class distributions change, using a single fixed $\tau$ causes overall adjustment to conflict with new class-level corrections, directly harming final accuracy.
>
> **Updating only the overall adjustment while ignoring de-duplicated estimation is also detrimental.**
> In Table 4, under all $10$ settings on CIFAR-10-LT and CIFAR-100-LT, "w/o D-L" consistently underperforms "w/ D-L" (DDDE + LMC). For instance, under the reversed setting $(k _ {\max},k _ {\min})=(10,1)$ on CIFAR-10-LT, "w/ D-L" achieves $2.07\\%$ higher accuracy than "w/o D-L."
> These results show that when class-level estimation is unreliable, the LMC-learned $\tau$ becomes misguided, indicating that the interaction between the two adjustments is bidirectional.
> We have emphasized this interplay more clearly in the obvervations of Table 4 (Section 6.3 of the revised version).
>
> **W3: Some statements lack evidence or reference, e.g. line 085. "current work" embraces all works, see also 1. (this is not saying that any of [2-5] does).**
>
> **Response:** Thank you for your careful review of the paper. In addition to adding the missing references [2–5], we have explicitly restricted our claim to LA-based LTSSL methods around Line 085.

---

> > ### Author Response · Authors · 2025-11-20
> > **Response to W4**
> >
> > **W4 (Q2): The protocol (for CIFAR) chosen according to Du et al. ICML 2024 (Simpro) deviates from other, previously published protocols, e.g. from the cited paper [6], and makes comparisons difficult, in particular to state-of-the-art methods that remained un-cited and that use those previous protocols. What results are obtained with the protocol from Kim et al. 2020 or is there some other way to make the results comparable?**
> >
> > **Response:** We understand the concern regarding comparison protocols. We choose the Unified SSL Benchmark (USB) [7], because it provides a standardized codebase, unified backbone/hyperparameters, and consistent dataset splits, ensuring a strictly fair comparison, which is often difficult with varied legacy protocols.
> >
> > Regarding the results under the protocol of Kim et al. [6]: Since porting our method and all recent baselines back to the legacy protocol may introduce implementation inconsistencies, we instead re-implement the missing baselines [2-5] within the rigorous USB protocol.
> >
> > The experimental results under a total of $10$ settings on CIFAR-10-LT and CIFAR-100-LT are shown in the two tables below.
> > Here, we set $(N _ 1,M _ {k _ {\max}})=(1500,3000)$ and $(N _ 1,M _ {k _ {\max}})=(150,300)$ on CIFAR-10-LT and CIFAR-100-LT, respectively.
> > '$\sim$' in the $(k _ {\max},k _ {\min})$ row represents an arbitrary class.
> > From the results in the table, it can be observed that our CoLA achieves the highest accuracy, outperforming other methods.
> >
> > | Dataset               |             |               | CIFAR-10-LT |             |             |
> > | --------------        | ----------- | ------------- | ----------- | ----------- | ----------- |
> > | $(k _ {\max},k _ {\min})$   | $(1,10)$    | $(\sim,\sim)$ | $(10,1)$    | $(5,10)$    | $(10,5)$    |
> > | $(\gamma _ l,\gamma _ u)$ | $(100,100)$ | $(100,1)$     | $(100,100)$ | $(100,100)$ | $(100,100)$ |
> > | UDAL [2]              | $82.70\pm0.23$ | $80.09\pm0.31$ | $78.90\pm0.66$ | $79.48\pm1.51$ | $79.54\pm0.52$ |
> > | SoftMatch [3]         | $79.64\pm0.20$ | $80.31\pm0.33$ | $77.56\pm0.80$ | $78.96\pm1.52$ | $79.71\pm0.41$ |
> > | ADELLO [4]            | $\underline{83.23\pm0.30}$ | $\underline{80.88\pm0.42}$ | $\underline{80.12\pm0.40}$ | $\underline{79.61\pm1.26}$ | $\underline{80.30\pm0.68}$ |
> > | SICSSL [5]            | $76.98\pm0.45$ | $75.74\pm0.43$ | $76.20\pm0.80$ | $74.51\pm1.45$ | $75.11\pm0.67$ |
> > | CoLA (Ours)           | $\pmb{85.04\pm0.32}$ | $\pmb{84.83\pm0.34}$ | $\pmb{86.84\pm0.73}$ | $\pmb{85.16\pm0.73}$ | $\pmb{84.42\pm0.45}$ |
> > |   |   |   |   |   |   |
> >
> > | Dataset               |           |               | CIFAR-100-LT |            |            |
> > | --------------        | --------- | ------------- | ------------ | ---------- | ---------- |
> > | $(k _ {\max},k _ {\min})$   | $(1,100)$ | $(\sim,\sim)$ | $(100,1)$    | $(50,100)$ | $(100,50)$ |
> > | $(\gamma _ l,\gamma _ u)$ | $(10,10)$ | $(10,1)$      | $(10,10)$    | $(10,10)$  | $(10,10)$  |
> > | UDAL [2]              | $58.86\pm0.45$ | $48.62\pm0.93$ | $\underline{58.61\pm0.86}$ | $58.51\pm0.63$ | $59.27\pm0.77$ |
> > | SoftMatch [3]         | $57.46\pm0.50$ | $\underline{49.08\pm0.82}$ | $58.05\pm0.92$ | $57.18\pm0.81$ | $58.77\pm0.89$ |
> > | ADELLO [4]            | $\underline{59.16\pm0.44}$ | $48.88\pm0.97$ | $58.37\pm0.83$ | $\underline{59.04\pm0.76}$ | $\underline{59.31\pm1.01}$ |
> > | SICSSL [5]            | $54.69\pm0.53$ | $47.70\pm0.90$ | $55.87\pm1.02$ | $55.89\pm1.11$ | $56.55\pm1.22$ |
> > | CoLA (Ours)           | $\pmb{60.42\pm0.60}$ | $\pmb{51.28\pm0.45}$ | $\pmb{61.40\pm0.65}$ | $\pmb{60.10\pm0.41}$ | $\pmb{61.17\pm0.56}$ |
> > |   |   |   |   |   |   |

---

> > > ### Author Response · Authors · 2025-11-20
> > > **Response to W5**
> > >
> > > **W5 (Q3): For experiments that use compatible protocols, e.g. STL-10-LT, the proposed method is inferior to e.g. [4] and rather en par with [3]. As the statement about state-of-the-art results need to be revised: in which situation does the proposed method shows its main strengths and weaknesses?**
> > >
> > > **Response:** We respectfully clarify the performance comparison. The perceived inferiority stems from differing implementation details in the original papers:
> > > 1. Backbone consistency: ADELLO [4] incorporates low-confidence sample distillation within the backbone FixMatch. To measure the core contribution of our logit adjustment strategy, we compare all methods using the standard FixMatch as the backbone without auxiliary distillation tricks. The ablation experiments in paper ADELLO [4] also confirms a significant performance drop without distillation tricks. For example, on the STL-10 with $\gamma _ l=20$, the accuracy decreases from $74.6\\%$ to $67.1\\%$ when distillation tricks are not used.
> > >
> > > 2. Robust Evaluation: ADELLO reports averages over $3$ runs, while we report over $5$ runs for greater statistical significance.
> > >
> > > When compared under the identical experimental setting (standard FixMatch backbone, $5$ runs) on STL-10 (table below, where '$-$' in the $(\gamma _ l,\gamma _ u)$ row indicates that $\gamma _ u$ is unknown), CoLA consistently outperforms ADELLO and other competitors.
> > >
> > > | Dataset  |   | STL-10 |   |   |
> > > | --------------        | ------------------- | -------- | ------------------- | -------- |
> > > | $(N _ 1,M)$             | $(150,\approx100k)$ |          | $(450,\approx100k)$ |          |
> > > | $(\gamma _ l,\gamma _ u)$ | $(10,-)$            | $(20,-)$ | $(10,-)$            | $(20,-)$ |
> > > | UDAL [2]              | $69.13\pm0.90$ | $64.26\pm1.42$ | $77.26\pm1.04$ | $73.47\pm1.15$ |
> > > | SoftMatch [3]         | $\underline{71.58\pm0.42}$ | $\underline{67.84\pm0.36}$ | $\underline{78.58\pm0.41}$ | $\underline{75.98\pm0.81}$ |
> > > | ADELLO [4]            | $71.26\pm0.94$ | $67.36\pm1.37$ | $77.76\pm0.95$ | $75.17\pm1.06$ |
> > > | SICSSL [5]            | $67.33\pm1.73$ | $58.03\pm1.39$ | $72.99\pm0.94$ | $68.21\pm1.71$ |
> > > | CoLA (Ours)           | $\pmb{73.32\pm0.73}$ | $\pmb{68.96\pm2.55}$ | $\pmb{79.70\pm0.48}$ | $\pmb{77.53\pm0.80}$ |
> > > |   |   |   |   |   |

---

> > > > ### Author Response · Authors · 2025-11-20
> > > > **Response to W6**
> > > >
> > > > **W6 (Q4): The description of erank is not sufficiently self-contained. In particular, it is not clear why the EN is estimated by erank. Line 197 just says "we quantify EN ... using ... erank" and references point to EN and erank, but not why the quantification is possible. Also, it is not obvious why line 205 $p(i)$ is a probability and not just a point in the $m _ y$-simplex. Why is the quantification possible and why is $p(i)$ a proper probability?**
> > > >
> > > > **Response:** Thank you for your valuable feedback. We will address your questions one by one.
> > > >
> > > > **1. Why is the quantification possible?**
> > > >
> > > > We apologize for the logical leap in Section 4.1. The detailed logical chain is as follows: as illustrated in Figure 1(a) of the introduction and described in the methodology, the head class has a large sample size but often comes with a high degree of sample redundancy. Simple frequency counting would overstate the importance of the head class due to these redundant samples, leading to the over-suppression problem. Therefore, we need an indicator to measure the number of "truly effective" or "informatively valuable" samples in a class, i.e., the number of samples after removing redundancy. This is the original intention of our reference to the concept of Effective Number (EN). Effective rank (erank) is precisely the standard tool for measuring this kind of feature-level diversity.
> > > >
> > > > We first collect high-confidence feature vectors for each class $y$, forming the feature matrix $\pmb{Z} _ y$. Performing singular value decomposition (SVD) on $\pmb{Z} _ y$, the resulting singular values $s _ i$ represent the energy of the data along each principal component. If a class has highly redundant samples (e.g., very similar images), its feature vectors $\pmb{z}^y _ j$ will be concentrated in a low-dimensional subspace. This leads to a very spiky singular value distribution, where most of the energy is concentrated in a few singular values; if a class has highly diverse samples, its feature vectors will be distributed in a higher-dimensional space. This results in a relatively flat singular value distribution, where the energy is more evenly spread across multiple singular values.
> > > >
> > > > The erank is defined as the exponentiated Shannon entropy of the singular value distribution $\exp\left(-\sum p(i)\log p(i)\right)$. Shannon entropy itself measures the flatness (uncertainty) of a distribution. When the distribution is spiky (highly redundant), the entropy is very low, and the erank value is also low; when the distribution is flat (highly diverse), the entropy is very high, and the erank value is also high.
> > > > Thus, erank directly quantifies the effective dimensionality or feature richness of the matrix $\pmb{Z} _ y$, and can serve as a proxy for EN.
> > > > We have supplemented the reason for using erank as a proxy for EN in Section 4.1.
> > > >
> > > > **2. Why is $p(i)$ a probability not a point in the $m _ y$-simplex?**
> > > >
> > > > $p(i)$ is a valid probability distribution based on the following two conditions:
> > > > (1) Non-negativity: According to the definition of singular value decomposition, all singular values $s _ i$ of matrix $\pmb{Z} _ y$ are non-negative ($s _ i \ge 0$). Therefore, their sum $\sum^{m _ y} _ {j=1} s _ j$ is also non-negative (in our code, if the sum is $0$, we return erank=$0$, so the sum involved in subsequent calculations is strictly positive). Thus, for all $i$, $p(i) = s _ i / \sum^{m _ y} _ {j=1} s _ j \ge 0$.
> > > > (2) Sum-to-one: The sum of all $p(i)$ equals $1$:
> > > >
> > > > $$\sum^{m _ y} _ {i=1} p(i) = \sum^{m _ y} _ {i=1} \left( \frac{s _ i}{\sum^{m _ y} _ {j=1} s _ j} \right) = \frac{1}{\sum^{m _ y} _ {j=1} s _ j} \sum^{m _ y} _ {i=1} s _ i = 1.$$
> > > >
> > > > Since $p(i)$ satisfies non-negativity and sums to $1$, it strictly defines a discrete probability distribution over $m _ y$ singular values. The vector $(p(1),...,p(m _ y))$ is a point in the $m _ y$-simplex space.
> > > >
> > > > **3. Why is $p(i)$ a proper probability?**
> > > >
> > > > As mentioned in the manuscript, the singular values $s _ i$ represent the energy of the data along its principal components. Therefore, $p(i)$ can be physically interpreted as the percentage of energy accounted for by the $i$-th principal component. We normalize the singular values in this form to legitimately apply Shannon entropy. Shannon entropy is an information-theoretic measure defined on probability distributions. Only by computing the entropy of this energy distribution can we measure the concentration or dispersion of energy, thereby quantifying the diversity of features.

---

> > > > > ### Author Response · Authors · 2025-11-20
> > > > > **Response to W7**
> > > > >
> > > > > **W7 (Q5): 4.2 is partly written in a procedural way and the overall approach that leads to the algorithm needs to be stated more clearly.**
> > > > >
> > > > > **Response:** Thanks for your suggestion. We have provided the pseudocode in the revised manuscript (Page 21, Algorithm 1) and have refined the overview paragraph at the beginning of Section 4 to provide a clear procedural reference.
> > > > >
> > > > > References:
> > > > > [1] Zhang et al., Mixup: Beyond empirical risk minimization. ICLR 2018.
> > > > > [2] Lazarow et al., Unifying distribution alignment as a loss for imbalanced semi-supervised learning. WACV 2023.
> > > > > [3] Chen et al., Softmatch: Addressing the quantity-quality tradeoff in semi-supervised learning. ICLR 2023.
> > > > > [4] Aimar et al., Flexible distribution alignment: Towards long-tailed semi-supervised learning with proper calibration. ECCV 2024.
> > > > > [5] Kim et al., Separated and Independent Contrastive Semi-Supervised Learning for Imbalanced Datasets. IEEE Access 2025.
> > > > > [6] Kim et al., Distribution aligning refinery of pseudo-label for imbalanced semi-supervised learning. NeurIPS 2020.
> > > > > [7] Wang et al., USB: A Unified Semi-supervised Learning Benchmark for Classification. NeurIPS Datasets and Benchmarks 2022.

---

> ### Comment · Reviewer_bnRy · 2025-11-20
> **Follow-up on W6 response 2./3.**
>
> Properties (1) and (2) define a point in the simplex. This is necessary but not sufficient to call $p(i)$ a probability (prediction). For the latter to apply, it is important to explain which random variables are occurring and what is estimated. The authors argument 3. makes an attempt in this direction, but some version of this also needs to be added to the manuscript.

---

> > ### Author Response · Authors · 2025-11-21
> > **Response to the probability question**
> >
> > Thank you for this insightful comment. We completely agree that satisfying the simplex constraints (non-negativity and sum-to-one) is necessary but not sufficient to define a probability distribution. A proper probabilistic interpretation requires a clear definition of the underlying random variable and the sample space. To address this, we clarify the probabilistic framework underlying the Effective Rank (erank) as follows:
> >
> > First, we consider **a discrete random variable, denoted as $\mathcal{K}$**, which represents the index of the principal component or the mode of variation in the feature space defined by $\pmb{Z} _ y$. The sample space of $\mathcal{K}$ is the set of indices $\\{1, 2, ..., m _ y\\}$.
> >
> > Then the normalized singular values $p(i)$ estimate **the probability mass function of $\mathcal{K}$**. Specifically, in the context of Principal Component Analysis (PCA) and spectral theory, the magnitude of a singular value $s _ i$ corresponds to the amount of variance or energy explained by the $i$-th principal direction. Therefore, $p(i) = s _ i / \sum _ j s _ j$ represents the probability that a random unit of feature variance or signal energy is aligned with the $i$-th principal component.
> >
> > Under this interpretation, calculating the Shannon entropy of $p(i)$ is mathematically rigorous. It measures the uncertainty of the random variable $\mathcal{K}$.
> >
> > We have revised Section 4.1 of the manuscript to explicitly include this definition of the random variable and the probabilistic interpretation of the singular value spectrum.

---

> > > ### Comment · Reviewer_bnRy · 2025-11-24
> > > **Follow-up on WP5 and WP6**
> > >
> > > While the authors' follow-up response to WP6 is satisfying, the reviewer disagrees with the argument regarding W5 (Q3) and "auxiliary distillation tricks":
> > > Similar to ACR and CDMAP, ADELLO suggests a "trick" (low-confidence distillation-based regularization) as one of its contributions. It is fine to compare the proposed method with existing methods without those "tricks" in the ablations - but then this needs to be done consistently with all such methods, including ACR. Whether calling these "tricks" is a separate question.
> > >
> > > However, for the SOTA comparisons, the full set of contributions of the existing methods must be taken into account. In case that the proposed method stays below SOTA, the reasons should be discussed, e.g. referring to the ablation studies and the findings regarding the additional steps.

---

> > > > ### Author Response · Authors · 2025-11-25
> > > > **Clarification on ADELLO's Distillation and Dataset Characteristics**
> > > >
> > > > We thank the reviewer for pushing us to clarify the comparison with ADELLO. We acknowledge that for a *System-Level SOTA* claim, the full ADELLO method should be considered. However, we would like to offer a deeper scientific explanation for the performance differences on STL-10, which supports the orthogonality of our contributions.
> > > >
> > > > We re-evaluate the performance of CoLA versus the full ADELLO, with experimental results shown in the table below. It can be observed that:
> > > > (1) The full ADELLO outperforms CoLA under the first, second, and fourth settings (columns), which may be because the STL-10 dataset contains a large number of out-of-distribution (OOD) samples in the unlabeled set, and the low-confidence distillation in ADELLO effectively acts as a threshold-based OOD filtering or regularization mechanism. It prevents the model from confidently and incorrectly assigning OOD samples to labeled categories or leveraging them for representation learning without hard label assignments, which is particularly advantageous on STL-10.
> > > > (2) When ADELLO does not employ distillation techniques (i.e., ADELLO*), focusing the comparison on handling imbalance, CoLA significantly outperforms ADELLO. This demonstrates that CoLA offers a superior solution for the core long-tail problem.
> > > > The contributions of these two approaches are largely orthogonal.
> > > >
> > > > | Dataset  |   | STL-10 |   |   |
> > > > | --------------        | ------------------- | -------- | ------------------- | -------- |
> > > > | $(N_1,M)$             | $(150,\approx100k)$ |          | $(450,\approx100k)$ |          |
> > > > | $(\gamma_l,\gamma_u)$ | $(10,-)$            | $(20,-)$ | $(10,-)$            | $(20,-)$ |
> > > > | ADELLO*     | $71.26\pm0.94$ | $67.36\pm1.37$ | $77.76\pm0.95$ | $75.17\pm1.06$ |
> > > > | ADELLO      | $\pmb{74.24\pm0.65}$ | $\pmb{71.56\pm0.74}$ | $\underline{79.44\pm0.46}$ | $\pmb{77.95\pm0.73}$ |
> > > > | CoLA (Ours) | $\underline{73.32\pm0.73}$ | $\underline{68.96\pm2.55}$ | $\pmb{79.70\pm0.48}$ | $\underline{77.53\pm0.80}$ |
> > > > |   |   |   |   |   |
> > > > '$*$' represents w/o distillation.
> > > >
> > > > We have added the above experimental results and discussion to Section 6.2.2. Additionally, we update the statements in the abstract (Line 30), introduction (Line 97) and conclusion (Line 515) to reflect that CoLA achieves SOTA in the standard long-tail setup and is highly competitive in the OOD scenario (STL-10) without requiring auxiliary distillation.

---

> > > > > ### Comment · Reviewer_bnRy · 2025-11-27
> > > > > **ACR and CDMAP**
> > > > >
> > > > > The table in the response is very useful, but covers the comments only partly. How about ACR and CDMAP, why are they not included? Also, what code-base/parameters have been used when values deviate from the reported in the literature?

---

> > > > > > ### Author Response · Authors · 2025-11-27
> > > > > > **Response to Reviewer bnRY**
> > > > > >
> > > > > > We thank the reviewer for the positive feedback on the additional analysis and assume the reviewer refers to CDMAD [1].
> > > > > >
> > > > > > We would like to clarify that the specific table in our previous response was designed as an ablation study specifically for the auxiliary distillation component unique to ADELLO, which effectively filters OOD samples on STL-10. We excluded ACR and CDMAD from that specific table because they do not employ this particular low-confidence distillation mechanism for OOD filtering, but rather focus on adaptive consistency regularization (ACR) or bias estimation (CDMAD). Therefore, comparing them in an ablation of distillation context was not applicable.
> > > > > >
> > > > > > Regarding the inclusion of ACR:
> > > > > > We respectfully note that ACR is already included in our original manuscript (Table 2). To facilitate a direct comparison as requested, we have added ACR to the table below. It can be observed that CoLA and ADELLO (Full) outperform ACR across almost all STL-10 settings.
> > > > > >
> > > > > > Regarding the inclusion of CDMAD:
> > > > > > We acknowledge this is a highly relevant and recent SOTA method. Since the open-sourced code of CDMAD is not based on the Unified SSL Benchmark (USB) [2], reproducing it strictly under our standardized setting requires significant adaptation. We will include the results of CDMAD in the final version.
> > > > > >
> > > > > > Regarding codebase and parameters:
> > > > > > The deviations from values reported in the literature stem from our rigorous standardization. We re-implement and train all methods (including ACR and ADELLO) using USB to ensure a fair comparison, rather than relying on numbers reported in papers. As detailed in Appendix G.2, we standardize these factors for all methods:
> > > > > > Backbone: Wide ResNet-28-2 (CIFAR/STL-10) and ResNet-50 (SIN-127).
> > > > > > Optimizer: SGD with momentum $0.9$.
> > > > > > Seed: Average over $5$ independent runs (seeds $0-4$).
> > > > > > Environment: PyTorch on NVIDIA RTX 4090 GPUs.
> > > > > >
> > > > > >
> > > > > >
> > > > > > | Dataset  |   | STL-10 |   |   |
> > > > > > | --------------        | ------------------- | -------- | ------------------- | -------- |
> > > > > > | $(N_1,M)$             | $(150,\approx100k)$ |          | $(450,\approx100k)$ |          |
> > > > > > | $(\gamma_l,\gamma_u)$ | $(10,-)$            | $(20,-)$ | $(10,-)$            | $(20,-)$ |
> > > > > > | ACR         | $71.26\pm0.75$ | $68.13\pm1.51$ | $79.26\pm0.44$ | $75.90\pm0.68$ |
> > > > > > | ADELLO      | $\pmb{74.24\pm0.65}$ | $\pmb{71.56\pm0.74}$ | $\underline{79.44\pm0.46}$ | $\pmb{77.95\pm0.73}$ |
> > > > > > | CoLA (Ours) | $\underline{73.32\pm0.73}$ | $\underline{68.96\pm2.55}$ | $\pmb{79.70\pm0.48}$ | $\underline{77.53\pm0.80}$ |
> > > > > > |   |   |   |   |   |
> > > > > >
> > > > > > [1] Lee et al., CDMAD: Class-Distribution-Mismatch-Aware Debiasing  for Class-Imbalanced Semi-Supervised Learning. CVPR 2024.
> > > > > > [2] Wang et al., USB: A Unified Semi-supervised Learning Benchmark for Classification. NeurIPS Datasets and Benchmarks 2022.

---

### Official Review · Reviewer_nQAr · 2025-11-03

**Soundness:** 2
**Presentation:** 3
**Contribution:** 3
**Rating:** 6
**Confidence:** 3

**Summary:**

This paper focuses on improving Long-Tailed Semi-Supervised Learning (LTSSL) by employing logit-adjustment methods. The paper identifies two key issues with the logit-adjustment approaches: 1) An over-estimation of popular class probabilities can lead to over-suppression of model predictions for the popular classes; and 2) The overall adjustment factor in the logit-adjustment approach is sensitive to particular dataset and needs to selected carefully. The paper then proposes two solutions to address these two issues, namely *de-duplicated distribution estimation* (DDDE) and *logit meta-calibration* (LMC). The paper provides a generalization analysis for LMC. The paper then provides a comprehensive empirical evidence of the value of the proposed approach while comparing it with existing logit-adjustment-based approaches and other methods beyond logit-adjustment for LTSSL in the literature.

**Strengths:**

- The paper studies a well motivated problem by identifying key limitations of a widely popular approach in the literature.
- The empirical results in the paper clearly showcase the improvements compared to competitive baselines from the literature.
- The paper presents ablation results to establish the value of both DDDE and LMC.

**Weaknesses:**

- The generalization analysis in Section 5 does not significantly enhance the overall contributions of the paper. Does the theory inspire/motivate the method proposed in the paper? If not, does the theory provide useful guarantees towards the final performance of the proposed solution?
- The presentation of the theoretical part of the paper can be greatly improved.
   -  Could the authors expand on the Line 298 (``If our estimation is accurate,...justifying our methodology``) and make it mathematically precise.
   - If the reviewer understand it correctly, the only optimizing parameter in Section 5 is $\tau$ and the function class $h_{\tau}$ is linear with respect to $\tau$ (since $\tau$ is the overall logit-adjustment factor). Could the author attempt to provide a more explicit characterization of the Rademarcher complexity in this case?

**Questions:**

- Could you please provide a description/pseudocode of your overall method in the form of an algorithmic block or a figure?
- In Line 275, the assumption is that $P\_{X\_{u} \mid Y\_{u}}(\mathbf{x}|y) =  P\_{X\_{l} \mid Y\_{l}}(\mathbf{x}|y)$ (between $u$ and $l$). However, the importance weight in 277 deals with $P\_{X\_{u}, Y\_{u}}$ and  $P\_{X\_{v}, Y\_{v}}$. Are the authors relying on the fact that since $\mathcal{D}\_{v}$ is sub-sampled from $\mathcal{D}\_{l}$ and thus share the same class conditionals? If yes, please consider making this clearer.
- Please consider making the notations consistent. E.g., Line 286 uses $R_{P\_u}$ (with lowercase $u$) while Line 294 uses $R_{P\_U}$ (with uppercase $U$).
- Did you use the logit-adjustment form in Line 234 (as opposed to the one in Eq. (1)) for your experiments?
- Why have you used the form in Line 226 for sampling? Why can one not simply use $\hat{P}\_{Y\_u}(y\_i)$ as the probability to select $(\mathbf{x}^l\_i, y\_i)$?

---

> ### Author Response · Authors · 2025-11-20
> **Responses to W1-W2**
>
> Thank you very much for your insightful comments, which have significantly helped us enhance the paper and highlight its contributions in a better way. We hope our responses can address all your concerns.
>
> **W1: The generalization analysis in Section 5 does not significantly enhance the overall contributions of the paper. Does the theory inspire/motivate the method proposed in the paper? If not, does the theory provide useful guarantees towards the final performance of the proposed solution?**
>
> **Response:** Thank you for this insightful question regarding the connection between our theory (Section 5) and our method (Section 4). The generalization analysis is indeed foundational to our paper, serving both to **motivate** the LMC procedure itself and to **provide useful guarantees** for its components.
>
> First, Proposition 1 (Section 5) provides the central theoretical motivation for this entire strategy. It explicitly bounds the target risk $R _ {P _ u}$ by the proxy risk $\hat{R} _ {\mathcal{D} _ v}$ plus other terms. This bound formally justifies that minimizing our LMC objective $\hat{R} _ {\mathcal{D} _ v}$ is a principled approach to minimizing the true target risk $R _ {P _ u}$. Without this bound, the LMC procedure would lack theoretical justification.
>
> Beyond motivating the LMC objective, the theory also provides two critical, non-trivial guarantees about why our proposed solution is effective:
> 1. It justifies the necessity of DDDE (Section 4.1). The bound's tightness depends directly on the discrepancy term $|\hat{R} _ {\mathcal{D} _ v,w}-\hat{R} _ {\mathcal{D} _ v}|$. This term measures the divergence between the proxy distribution $P _ v$ and the target distribution $P _ u$. By our construction (Assumption 2), $P _ {Y _ v}(y) = \hat{P} _ {Y _ u}(y)$, and under Assumption 3, the term simplifies to reflect the accuracy of our estimate $\hat{P} _ {Y _ u}(y)$ (via the weight $w(y) = P _ {Y _ u}(y) / \hat{P} _ {Y _ u}(y)$).
> This theoretically demonstrates why our DDDE method is crucial for the success of LMC. **A more accurate distribution estimate (as provided by DDDE) leads to a smaller discrepancy term, a tighter bound, and thus a more reliable $\tau^*$ found by LMC. This links the two components of our method together.**
> 2. It inspires the convex formulation of LMC. Our theoretical formulation also directly inspired our specific design choice of using the linear adjustment term ($-\tau\cdot\pmb{p}$) over the traditional log-term ($-\tau\cdot\log\pmb{p}$), as noted in Section 4.2. As we analyze in Appendix F, this linear formulation results in the LMC objective function $\mathcal{L}(\tau)$ being convex with respect to $\tau$. This is a powerful and highly useful guarantee for the final performance. It ensures that our optimization for $\tau$ is robust, efficient, and is guaranteed to find the global optimum $\tau^*$, which would not be true for a non-convex objective.
>
> In summary, the generalization analysis is not an isolated component. It provides the core motivation for using a proxy set (LMC), justifies the need for high-quality distribution estimation (DDDE), and inspires the convex formulation that guarantees an optimal and robust solution.
>
>
>
> **W2: Could the authors expand on the Line 298 (If our estimation is accurate,...justifying our methodology) and make it mathematically precise.**
>
> **Response:** Thank you for your suggestion. Specifically, the discrepancy term can be controlled via the class-proportion error:
> $$
> \left|\hat{R} _ {\mathcal{D} _ v,w}(h _ \tau)-\hat{R} _ {\mathcal{D} _ v}(h _ \tau)\right|=\left|\frac{1}{V}\sum _ {i=1}^{V}\big(w(y _ i^v)-1\big)\\,\ell(h _ \tau(\pmb{x} _ i^v),y _ i^v)\right|\le \frac{U}{V}\sum _ {i=1}^{V}\big|w(y _ i^v)-1\big|.
> $$
> Because $w(y)=\frac{P _ {Y _ u}(y)}{P _ {Y _ v}(y)}$ and $P _ {Y _ v}(y)=\hat{P} _ {Y _ u}(y)$ by Assumption 2, taking expectations over $\mathcal{D} _ v$ gives
> $$
> \mathbb{E} _ {\mathcal{D} _ v}\\!\left[ \frac{U}{V}\sum _ {i=1}^{V}\big|w(y _ i^v)-1\big| \right] = U\sum _ {y\in\mathcal{Y}}\big|P _ {Y _ u}(y)-\hat{P} _ {Y _ u}(y)\big|=U\\,\big\\|P _ {Y _ u}-\hat{P} _ {Y _ u}\big\\| _ 1.
> $$
>
> Moreover, since $0 \le w(y) \le B$ (Assumption 4), the term $|w(y _ i^v)-1|$ is bounded by $\max(1, B-1)$.
> By applying Hoeffding’s inequality to the empirical mean $\frac{1}{V}\sum _ i^V U \cdot |w(y _ i^v)-1|$, we have that with probability at least $1-\delta$:
> $$
> \left|\hat{R} _ {\mathcal{D} _ v,w}(h _ \tau)-\hat{R} _ {\mathcal{D} _ v}(h _ \tau)\right| \le U\left(\big\\|P _ {Y _ u}-\hat{P} _ {Y _ u}\big\\| _ 1 + \max(1, B-1)\sqrt{\frac{\log(1/\delta)}{2V}}\right).
> $$
> Thus, when our estimator $\hat{P} _ {Y _ u}$ is accurate (small $\ell _ 1$ error), the discrepancy term contracts at the same rate, formally tightening the overall bound and validating the methodology. We have added the above content to Appendix E.

---

> > ### Author Response · Authors · 2025-11-20
> > **Response to W3**
> >
> > **W3: If the reviewer understands it correctly, the only optimizing parameter in Section 5 is $\tau$ and the function class $h _ {\tau}$ is linear with respect to $\tau$ (since $\tau$ is the overall logit-adjustment factor). Could the author attempt to provide a more explicit characterization of the Rademacher complexity in this case?**
> >
> > **Response:** This is a good point. We provide the full derivation below.
> >
> > **Setup and Notation**
> > Denote the backbone logits (kept fixed when optimizing $\tau$) by $s(\pmb{x})\in\mathbb{R}^K$ and write $s _ y(\pmb{x})$ for the $y$-th component. Our LMC applies a linear correction $-\tau\\,\pmb{p}$ with $\pmb{p}=(\hat{P} _ {Y _ u}(1),\dots,\hat{P} _ {Y _ u}(K))$. Hence
> > $$
> > h _ \tau(\pmb{x}) _ y = s _ y(\pmb{x})-\tau\\, p _ y, \qquad p _ y := \hat{P} _ {Y _ u}(y)\ge 0.
> > $$
> > The relevant real-valued hypothesis class is therefore
> > $$
> >     \mathcal{H} _ \tau = \big\\{ (\pmb{x},y) \mapsto s _ y(\pmb{x}) - \tau\\,p _ y \\,\big|\\, \tau\in\mathcal{T} \big\\},
> > $$
> > where $\mathcal{T}=[0,\tau _ {\max}]$ is the search interval used in practice (without a bounded interval, the complexity would be unbounded).
> >
> > **Empirical Rademacher Complexity**
> > For $\mathcal{D} _ v$, the empirical complexity satisfies
> > $$
> >     \widehat{\mathfrak{R}} _ {\mathcal{D} _ v}(\mathcal{H} _ \tau)
> >     = \mathbb{E} _ {\pmb{\sigma}}\left[\sup _ {\tau\in\mathcal{T}}\frac{1}{V}\sum _ {i=1}^V \sigma _ i\big(s _ {y _ i^v}(\pmb{x} _ i^v)-\tau\\,p _ {y _ i^v}\big)\right],
> > $$
> > with $\sigma _ i\in\\{-1,+1\\}$. Because the logits are constant with respect to $\tau$, this splits into
> > \begin{align*}
> >     \widehat{\mathfrak{R}} _ {\mathcal{D} _ v}(\mathcal{H} _ \tau)
> >     & = \frac{1}{V}\mathbb{E} _ {\pmb{\sigma}}\left[\sum _ {i=1}^V \sigma _ i s _ {y _ i^v}(\pmb{x} _ i^v)\right]
> >     + \frac{1}{V}\mathbb{E} _ {\pmb{\sigma}}\left[\sup _ {\tau\in[0,\tau _ {\max}]}\left(-\tau\sum _ {i=1}^V \sigma _ i p _ {y _ i^v}\right)\right]\\\\
> >     & = \frac{1}{V}\mathbb{E} _ {\pmb{\sigma}}\left[\sup _ {\tau\in[0,\tau _ {\max}]}\left(-\tau\sum _ {i=1}^V \sigma _ i p _ {y _ i^v}\right)\right],
> > \end{align*}
> > because $\mathbb{E}[\sigma _ i]=0$. Let $S _ p:=\sum _ {i=1}^V\sigma _ i p _ {y _ i^v}$. If $S _ p\ge 0$ the supremum is achieved at $\tau = 0$; otherwise it is achieved at $\tau _ {\max}$, yielding $\tau _ {\max}|S _ p|$. Consequently
> > $$
> >     \widehat{\mathfrak{R}} _ {\mathcal{D} _ v}(\mathcal{H} _ \tau)
> >     \le \frac{\tau _ {\max}}{V}\\,\mathbb{E} _ {\pmb{\sigma}}\\!\left[|S _ p|\right].
> >     \tag{1}
> > $$
> >
> >
> >
> > **Controlling the Rademacher Sum**
> > Let $n _ y$ be the count of class $y$ inside $\mathcal{D} _ v$. Then
> > $$
> >     S _ p = \sum _ {y\in\mathcal{Y}} p _ y \left(\sum _ {i:y _ i^v=y}\sigma _ i\right)
> >     \quad\text{and}\quad
> >     \mathbb{E} _ {\pmb{\sigma}}\\!\left[S _ p^2\right] = \sum _ {y\in\mathcal{Y}} n _ y p _ y^2.
> > $$
> > By Khintchine's inequality,
> > $$
> >     \mathbb{E} _ {\pmb{\sigma}}\\!\left[|S _ p|\right] \le \sqrt{\mathbb{E} _ {\pmb{\sigma}}\left[S _ p^2\right]}
> >     = \sqrt{\sum _ {y\in\mathcal{Y}} n _ y p _ y^2}.
> >     \tag{2}
> > $$
> > Combining $(1)$ and $(2)$ yields the finite-sample bound
> > $$
> >     \widehat{\mathfrak{R}} _ {\mathcal{D} _ v}(\mathcal{H} _ \tau)
> >     \le \frac{\tau _ {\max}}{V}\sqrt{\sum _ {y\in\mathcal{Y}} n _ y p _ y^2}.
> > $$
> > Taking expectation over draws of $\mathcal{D} _ v$ and applying Jensen's inequality to $\mathbb{E}\big[\sqrt{\sum _ {y\in\mathcal{Y}} n _ y p _ y^2}\big]$ yields
> > $$
> >     \mathfrak{R} _ V(\mathcal{H} _ \tau)
> >     \le \frac{\tau _ {\max}}{\sqrt{V}}\sqrt{\sum _ {y\in\mathcal{Y}} p _ y^3}.
> >     \tag{3}
> > $$
> >
> > **Plugging Back into Proposition 1**
> > Using $(3)$ inside Proposition 1 leads to
> > $$
> >     R _ {P _ u}(h _ \tau)
> >     \le \hat{R} _ {\mathcal{D} _ v}(h _ \tau)
> >     + \big|\hat{R} _ {\mathcal{D} _ v,w}(h _ \tau) - \hat{R} _ {\mathcal{D} _ v}(h _ \tau)\big| + \frac{2 B L \tau _ {\max}}{\sqrt{V}}
> >     \sqrt{\sum _ {y\in\mathcal{Y}} p _ y^3}
> >     + U B \sqrt{\frac{\log(1/\delta)}{2V}}.
> > $$
> > From the above inequality, we can draw the following observations:
> > 1. The third term scales as $1/\sqrt{V}$ and depends only on the search radius $\tau _ {\max}$ and the geometry of the target class prior, not on the underlying deep backbone.
> > 2. The quantity $\sum _ {y\in\mathcal{Y}} p _ y^3$ (a third moment of the class prior) shrinks when the distribution is closer to uniform and remains bounded even under heavy imbalance.
> > 3. Without $\tau\in[0,\tau _ {\max}]$, $\mathfrak{R} _ V(\mathcal{H} _ \tau)$ would diverge. In practice the meta-search already constrains $\tau$, which simultaneously guarantees a finite complexity term.
> >
> > Consequently, the hypothesis class induced by meta-calibrating a single scalar admits the explicit control $(3)$, supplying the desired clarification for Proposition 1. We have added the above content to Appendix E.

---

> > > ### Author Response · Authors · 2025-11-20
> > > **Responses to Q1-Q5**
> > >
> > > **Q1: Could you please provide a description/pseudocode of your overall method in the form of an algorithmic block or a figure?**
> > >
> > > **Response:** We follow your suggestion and have provided the pseudocode in the revised manuscript (Page 21, Algorithm 1). We have also refined the overview paragraph at the beginning of Section 4 to provide a clear algorithmic workflow.
> > >
> > >
> > > **Q2: In Line 275, the assumption is that $P _ {\pmb{X} _ u|Y _ u}(\pmb{x}|y)=P _ {\pmb{X} _ l|Y _ l}(\pmb{x}|y)$ (between $u$ and $l$). However, the importance weight in 277 deals with $P _ {\pmb{X} _ u,Y _ u}$ and $P _ {\pmb{X} _ v,Y _ v}$. Are the authors relying on the fact that since $\mathcal{D} _ v$ is sub-sampled from $\mathcal{D} _ l$ and thus share the same class conditionals? If yes, please consider making this clearer.**
> > >
> > > **Response:** Thank you for your suggestion. You are absolutely correct. In our methodology (Section 4.2), the proxy set $\mathcal{D} _ v$ is constructed via rejection sampling from $\mathcal{D} _ l$. Crucially, the selection probability for each sample, as defined in Line 226, depends solely on its class label $y$ and the class frequencies, and is independent of the feature $\pmb{x}$.
> > > Statistically, this ensures that within each class, samples are drawn uniformly at random. Therefore, while the class priors $P _ {Y _ v}(y)$ are altered to match the target distribution $\hat{P} _ {Y _ u}(y)$, the class-conditional distribution remains invariant, i.e., $P _ {\pmb{X} _ v|Y _ v}(\pmb{x}|y) = P _ {\pmb{X} _ l|Y _ l}(\pmb{x}|y)$.
> > > We have already added supplementary explanations for this point in the revised version.
> > >
> > > **Q3: Please consider making the notations consistent. E.g., Line 286 uses $R _ {P _ u}$ (with lowercase $u$) while Line 294 uses $R _ {P _ U}$ (with uppercase $U$).**
> > >
> > > **Response:** We sincerely apologize for this oversight and thank the reviewer for the careful reading. We have corrected the typo from $R _ {P _ U}$ to $R _ {P _ u}$. We have also thoroughly proofread the entire manuscript to ensure notation consistency throughout.
> > >
> > >
> > >
> > > **Q4: Did you use the logit-adjustment form in Line 234 (as opposed to the one in Eq. (1)) for your experiments?**
> > >
> > > **Response:** Yes, that is correct. We intentionally adopt the linear adjustment form $-\tau \cdot \pmb{p}$ during and after LMC, rather than the logarithmic form used in standard post-hoc LA (Eq. (1)).
> > > As discussed in Section 4.2 and the response to Weakness 1, we choose this linear formulation for a more stable optimization process.
> > >
> > >
> > >
> > > **Q5: Why have you used the form in Line 226 for sampling? Why can one not simply use $\hat{P} _ {Y _ u}(y _ i)$ as the probability to select $(\pmb{x}^l _ i,y _ i)$?**
> > >
> > > **Response:** Thank you for raising this question regarding the sampling probability. The reason we use the form $\frac{\hat{P} _ {Y _ u}(y)}{N _ y}$ instead of simply $\hat{P} _ {Y _ u}(y)$ is that our source dataset $\mathcal{D} _ l$ is already imbalanced with varying sample counts $N _ y$ for each class.
> > > Our goal is to construct a validation set $\mathcal{D} _ v$ where the total number of samples for each class is proportional to the target distribution $\hat{P} _ {Y _ u}(y)$. The expected number of samples for class $y$ in $\mathcal{D} _ v$ is given by:
> > > $$
> > > \mathbb{E}[\text{count} _ y] = N _ y \times \mathbb{P}(\text{selecting a sample } (\pmb{x}, y)).
> > > $$
> > > With our formula ($\mathbb{P} \propto \frac{\hat{P} _ {Y _ u}(y)}{N _ y}$):
> > > $$
> > > \mathbb{E}[\text{count} _ y] \propto N _ y \times \frac{\hat{P} _ {Y _ u}(y)}{N _ y} = \hat{P} _ {Y _ u}(y).
> > > $$
> > > This successfully matches the target distribution. With the alternative ($\mathbb{P} \propto \hat{P} _ {Y _ u}(y)$):
> > > $$
> > > \mathbb{E}[\text{count} _ y] \propto N _ y \times \hat{P} _ {Y _ u}(y).
> > > $$
> > > This would result in a distribution that is the product of the original imbalance and the target distribution, failing to achieve the desired distribution match. The denominator term $\max _ {y\in\mathcal{Y}}(\hat{P} _ {Y _ u}(y)/N _ y)$ in Line 226 is simply a normalization constant to ensure the probabilities do not exceed $1$.

---

### Author Response · Authors · 2025-11-30
**Clarification on Score Reverting and Rebuttal Consensus**

We sincerely thank the Area Chair and all reviewers for their dedicated time and constructive feedback. We appreciate the opportunity to clarify our contributions, and we have incorporated all suggestions into the revised manuscript, including the pseudocode, complexity analysis, and extended benchmarks.

We urgently wish to inform the AC that a positive consensus was actively forming prior to the interruption of rebuttal, with the effective scores reaching 6/6/6/4.

**1. Confirmed Score Increase (Reviewer peuh: $4 \to 6$)**
Reviewer peuh explicitly acknowledged that our rebuttal resolved their concerns regarding novelty, computational cost, and the proxy set validity. In their final official comment, they stated: I have read the rebuttal and raise my score accordingly. This confirms that the true status of the paper includes at least three positive ratings.

**2. Positive Resolution with Reviewer bnRy (Score $4$)**
We engaged in deep, productive discussions with Reviewer bnRy regarding SOTA comparisons and the definition of erank.
The reviewer explicitly found our follow-up on erank **satisfying** and the new experimental tables **very useful**.
Due to an OpenReview bug, the discussion with bnRy ended prematurely. However, we believe these clarifications largely resolved their concerns.

**3. Consensus on Soundness (Reviewers nQAr & DkFy: Score $6$)**
Reviewers nQAr and DkFy rated the paper positively from the outset. We successfully addressed their specific technical requests:
Theory: Provided the complete derivation of Rademacher complexity and its link to the convex LMC objective for Reviewer nQAr.
Efficiency: Added detailed complexity analysis and runtime comparisons for Reviewer DkFy, proving CoLA introduces negligible overhead.

Given that Reviewer peuh has clearly improved the score and Reviewer bnRy has shown a positive attitude toward the discussion, we kindly request the AC to evaluate the paper based on the updated consensus (effectively 6/6/6/4) rather than the initial state.

---

### Meta-Review · Area_Chair_2vzk · 2026-01-06

**Summary:**

The submission studies long-tailed semi-supervised learning (LTSSL) and proposes a logit-adjustment-based framework (DDDE + LMC, instantiated as CoLA) aimed at mitigating pseudo-label bias under class-imbalance and distribution mismatch.

Reviewers generally agreed the problem is important and the empirical results are competitive on several benchmarks.
The rebuttal also substantially improved method clarity, added useful theoretical explanations and additional analysis.

However, some key issues remain unresolved for acceptance at this stage:

1. The novelty and positioning still read as an aggregation of known components rather than a clearly novel mechanism with unavoidable necessity.

2. The evaluation comparing CoLA with SOTA remains incomplete, leaving uncertainty about the method’s standing and the generality of the claims.

**Reviewer Concerns:**

Reviewers' concerns about the novelty, method description and computational overhead are well addressed in the rebuttal.

However, some concerns are only partly addressed in the rebuttal:

1. **System-level comparison with SOTA methods** raised by **reviewer bnRy**: The discussion acknowledges that some of the competing methods include additional components (e.g., ADELLO distillation) that can change outcomes. A strong and relevant baseline CDMAP is also missing for comparison. It remains unclear whether the proposed method is truly competitive in their setting.

2. **novelty issue** raised by **reviewer peuh**: While the rebuttal argues that each component has theoretical motivation, the overall method still appears to be a combination of known ideas. The theory additions mainly improve justification and clarity, but do not fully reduce the impression that the core advance is incremental relative to the breadth of prior LTSSL work.

3. **Effect of LMC on pseudo label accuracy** raised by **reviewer peuh**:
The rebuttal argues that pseudo-label accuracy slopes may not reflect generalization and points to ablations where LMC improves final accuracy. Still, the evidence that LMC is a decisive contributor remains somewhat indirect, reinforcing the broader “incremental composition” concern.

**Reviewer Scores:**

For **reviewer nQAr(score 6)** and **reviewer DkFy(score 6)**, they may keep their score to 6 as they didn't patriciate in the discussion.

For **reviewer peuh (score 4)**, he may increase his score to 6 because he explicitly mentions he would raise his scores after rebuttal.

For **reviewer bnRy (score 4)**, he may keep his score to 4 as his concerns are not fully addressed.

---

### Decision · Program_Chairs · 2026-01-26

Accept (Poster)